# A quasi-objective single buoy approach for understanding Lagrangian coherent structures and sea ice dynamics

Nikolas O. Aksamit[1, 2,†], Randall K. Scharien[1], Jennifer K. Hutchings[3], and Jennifer V. Lukovich[4]

[1]Department of Geography, University of Victoria, Canada
[2]School of Earth and Environment, University of Canterbury, New Zealand
[3]College of Earth, Ocean, and Atmospheric Sciences, Oregon State University, USA
[4]Centre for Earth Observation Science, University of Manitoba, Canada
[†]Now at Department of Mathematics and Statistics, UiT - The Arctic University of Norway, Norway

**Correspondence:** Nikolas O. Aksamit (nikolas.aksamit@uit.no)

**Abstract.** Sea ice drift and deformation, namely sea ice dynamics, play a significant role in atmosphere-ice-ocean coupling. Deformation patterns in sea ice can be observed over a wide range of spatial and temporal scales, though high resolution objective quantification of these features remains difficult. In an effort to better understand local deformation of sea ice, we adapt the Trajectory Stretching Exponents (TSEs), quasi-objective measures of Lagrangian stretching in continuous media, to sea ice buoy data, and develop a temporal analysis of TSE time series. Our work expands on previous ocean current studies that have shown TSEs provide an approximation of Lagrangian coherent structure diagnostics when only sparse trajectory data is available. As TSEs do not require multiple buoys, we find they have an expanded range of use when compared with traditional Eulerian buoy-array deformation metrics, and provide local-stretching information below the length-scales possible when averaging over buoy-arrays. We verify the ability of TSEs to temporally and spatially identify dynamic features for three different sea ice datasets. The ability of TSEs to quantify trajectory stretching is verified by concurrent ice fracture in buoy neighborhoods ranging from tens to hundreds of kilometers in diameter, as well as the temporal concurrence of significant storm events.

## 1 Introduction

The Arctic is warming at a much greater rate than the rest of the Earth and sea ice plays an important role in regulating energy exchanges between the atmosphere, cryosphere, and ocean. The recent decline in sea ice extent and the prevalence of younger, thinner ice is well documented (Rothrock et al., 1999; Kwok and Rothrock, 2009; Landy et al., 2022) with thin ice being more susceptible to deformation and fracture. As the ice warms in spring, melt is accelerated around existing fractures due to a reduction in albedo and the presence of more open water. Arctic amplification, the disproportionate warming of the Arctic in a changing global climate, has been partially attributed to the enhanced oceanic heating and ice-albedo feedback caused by diminishing sea ice (Screen and Simmonds, 2010; Dai et al., 2019; Thackeray and Hall, 2019; Jenkins and Dai, 2021). Changes in the Arctic sea ice cover in recent years have also led to modifications of larger atmospheric circulation patterns (Moore et al., 2018), and mid-latitude weather (Siew et al., 2020). It is therefore of great importance to accurately measure sea ice dynamics

to understand the state of ice deformation, fracture, and refreezing, as well as the impacts on global climate, infrastructure in Arctic waters, and access for northern communities.

Sea ice deformation is a physical phenomenon that is highly localized in space and time (Oikkonen et al., 2017; Rampal et al., 2019), with local and regional deformation influencing summer sea ice melt rates and future weather. Even short winter storms have been shown to have long-term impacts on sea ice extent and melt (Graham et al., 2019b; Lukovich et al., 2021). Quantitative methods that identify deformation rates and sea ice flow patterns are thus critical for understanding atmosphere-ice-ocean exchange processes and grounding sea ice models.

Previous studies have used both displacement grids and sea ice buoy trajectories to quantify observations of sea ice deformation (e.g., Rampal et al., 2009; Hutchings et al., 2011; Szanyi et al., 2016a; Oikkonen et al., 2017). Over several decades, various approaches have been developed, including both Eulerian diagnostics that focus on ice behavior in a single time slice and Lagrangian diagnostics that assess the evolution of ice flow and the underlying flow patterns (Szanyi et al., 2016a, b). Perhaps the most common sea ice deformation metrics are the instantaneous values divergence, shear, and total deformation as
defined from components of the two-dimensional velocity gradient, $\nabla \mathbf{v}$ (Leppäranta, 2011):

$$\text{div} = \frac{\partial v_1}{\partial x} + \frac{\partial v_2}{\partial y}. \tag{1}$$

$$\text{shr} = \sqrt{\left(\frac{\partial v_1}{\partial x} - \frac{\partial v_2}{\partial y}\right)^2 + \left(\frac{\partial v_1}{\partial y} + \frac{\partial v_2}{\partial x}\right)^2}, \tag{2}$$

$$D = \sqrt{\text{div}^2 + \text{shr}^2}. \tag{3}$$

These Eulerian diagnostics (1-3) have been used extensively for both sea ice model validation, and experimental observations.
An alternative approach to studying sea ice dynamics is through methods that robustly identify significant sea ice flow features that persist over time, i.e. Lagrangian coherent structures (LCSs). LCSs are rigorously defined mathematical structures adapted from dynamical systems theory that behave as the underlying skeleton of fluid flow, thus shaping patterns in the evolution of material (Haller, 2015). Of particular relevance to the study of sea ice deformation are hyperbolic LCS. As a material evolves over a particular time window, hyperbolic LCS identify the curves of maximum stretching and compression.
Previous work by Szanyi et al. (2016a, b) studied hyperbolic LCS to determine distinct zones of deformation in Arctic sea ice, and how they varied over time, but in general the extent of their use in sea ice dynamics is currently limited.

In Figure 1 we show an example of a hyperbolic LCS retrieved from ocean surface current data. This visualization reveals why hyperbolic LCS are particularly relevant for studying the deformation of sea ice. In panel 1a, an initial gray square of fluid at time $t_0$ is shown. The hyperbolic repelling ($\mathcal{M_R}$) and attracting ($\mathcal{M_A}$) manifolds are drawn in blue and red, respectively.
As the fluid evolves from time $t_0$ to $t_2$ (panels a-c), fluid particles are attracted to $\mathcal{M_A}$ with material stretching along that axis, and repel from $\mathcal{M_R}$, with compression along that direction. When evaluating sea ice dynamics from an LCS perspective, it

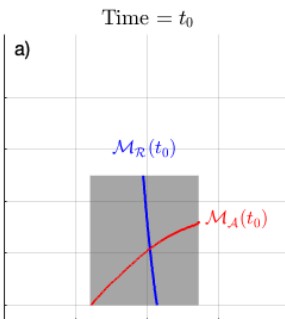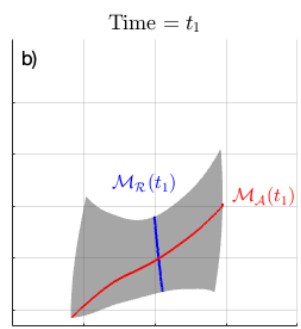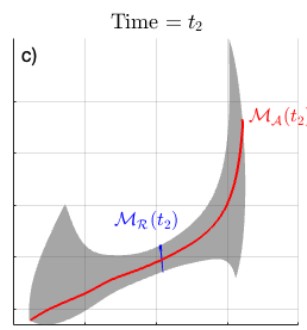

**Figure 1.** Example of time evolution of fluid particles surrounding hyperbolic LCSs in an incompressible ocean surface current simulation from time $t_0$ to $t_2$. $\mathcal{M}_\mathcal{A}$ is an attracting hyperbolic LCS (unstable manifold), and $\mathcal{M}_\mathcal{R}$ is a repelling hyperbolic LCS (stable manifold).

follows that separations between distinct dynamic regions may correspond with strong shear zones or separation features, such as cracks and ridges, as previously performed at a pan-Arctic scale by Szanyi et al. (2016a). It is our hypothesis that resolving the time-varying dynamics of these coherent structures may also help identify periods of significant ice modification, such as during influential storms or other breakup events.

Hyperbolic LCS are typically approximated by ridges of the finite time Lyapunov exponent (FTLE) (e.g., Szanyi et al., 2016a). Calculation of FTLE fields, however, rely on advecting grids of particles in time-resolved velocity fields and quantifying spatial gradients of the flow map, as discussed in detail in Appendix A1 and by Haller (2015). Obtaining gridded sea ice velocity data that can be used to calculate rate-of-strain invariants (1-3) and FTLE fields is a significant hurdle. This is currently only feasible at large scales via motion tracking algorithms and remotely sensed data. Several of these gridded velocity products are also known to be hindered by artificial velocity discontinuities caused when assimilating different data streams (Bouillon and Rampal, 2015; Szanyi et al., 2016b). These limitations prevent calculation of reliable Eulerian and Lagrangian diagnostics, especially at high (sub-daily) temporal resolution. In contrast, buoy GPS trajectories are often sampled at hourly or sub-hourly timescales and can provide a wealth of true sea ice motion data. Lagrangian and Eulerian diagnostics from these data sources can help address some of the limitations in gridded velocity studies.

Buoys passively follow ice floes, are typically sparse in the spatial domain, and we have limited control over where the data will be available. Therefore, spatial derivatives of ice velocity are not immediately available from buoy trajectories as with a gridded product. A common approach for obtaining (1-3) from buoys is to study arrays of buoys that form convex polygons, and calculate discretized contour integrals (via Green's theorem) at each timestep. This approach can be traced back to at least the early 1990's where it was applied to sparse sea ice motion vectors obtained from sequential SAR frames (Kwok et al., 1990). Deformation metrics can be calculated by tracking pre-defined polygons (Hutchings et al., 2011, 2012), or by looking

at statistics for all suitable triangles (e.g. no small angles) in an array (Itkin et al., 2017). For an $N$-sided polygon, with vertices indexed in counterclockwise orientation such that $(x_{N+1}, y_{N+1}) = (x_1, y_1)$, $\nabla \mathbf{v}$ can be approximated as follows:

$$
\begin{aligned}
\frac{\partial v_1}{\partial x} &= \frac{1}{2A} \sum_{i=1}^{N} [v_1(x_{i+1}, y_{i+1}) + v_1(x_i, y_i)][y_{i+1} - y_i], \\
\frac{\partial v_1}{\partial y} &= -\frac{1}{2A} \sum_{i=1}^{N} [v_1(x_{i+1}, y_{i+1}) + v_1(x_i, y_i)][x_{i+1} - x_i], \\
\frac{\partial v_2}{\partial x} &= \frac{1}{2A} \sum_{i=1}^{N} [v_2(x_{i+1}, y_{i+1}) + v_2(x_i, y_i)][y_{i+1} - y_i], \\
\frac{\partial v_2}{\partial y} &= -\frac{1}{2A} \sum_{i=1}^{N} [v_2(x_{i+1}, y_{i+1}) + v_2(x_i, y_i)][x_{i+1} - x_i]
\end{aligned}
\tag{4}
$$

where $A$ is the area of the polygon.

Complications with the Green's theorem method include sensitivities to user choice of array, accounting for GPS signal-to-noise ratios, and several physical inaccuracies arising from discretization. As arrays of buoys passively follow ice floes, they typically do not maintain a shape that allows accurate calculation of the spatial velocity gradients from drift trajectories. Some of these issues have been previously discussed by Lindsay and Stern (2003); Hutchings et al. (2012) and Dierking et al. (2020), including the polygon array selection method. Because of the necessary discretization of a continuous boundary, integral approximations by finite sums in the Green's theorem approach also generate errors. Lindsay and Stern (2003) previously addressed boundary representation errors for nonlinear velocity fields, but only errors from discontinuities (leads) that intersect polygon boundaries were studied. Uncertainty also stems from the trapezoid rule approximation of the contour integral (4). For a known continuous velocity field (e.g. without cracks), the upper bound of integral approximations can be quantified (Atkinson, 1989), though an explicit impact of this error cannot be found in the sea ice literature. As is detailed in Appendix A2, the trapezoid rule error can cause equilateral triads to indicate both divergent and convergent conditions in a steady, divergence-free, and continuous (fracture-free) flow, depending on array orientation.

In light of these complications, alternative methods that do not rely on buoy geometry or orientation, such as single trajectory metrics, could be largely beneficial, even if only to identify and separate dynamically active regions. One primary issue with single-trajectory metrics is that they are not frame-indifferent. That is, they depend on the reference frame of the observer. One may argue that it is sufficient to require all computations to be performed in the same reference frame, but one of the main axioms of mechanics is that the material response (e.g. ridging, fracturing) of a material is independent of the observer (Gurtin, 1981). One intuitive interpretation of this axiom is a fracture formed in ice is independent of whether the ice is viewed from an airplane, or from the ice surface. Frame-indifference is, therefore, a foundational benchmark that diagnostics must meet in order to identify coherent features in any deforming continuum. Invariants of the rate-of-strain tensor are frame-indifferent, as is FTLE, but velocity and velocity gradients are not.

As an alternative to Eulerian contour integral diagnostics, we propose using novel single-buoy stretching diagnostics, the trajectory stretching exponents (TSEs) (Haller et al., 2021, 2022), to approximate the influence of hyperbolic LCS. For a trajectory $\boldsymbol{x}(t)$, TSEs are quasi-objective Lagrangian metrics of material stretching. That is, they approximate true material stretching in slowly varying flows (i.e. when Lagrangian time scales dominate Eulerian time scales, as is typical in geophysical flows). As shown by Haller et al. (2021), TSEs identify the same hyperbolic LCS as FTLE in open ocean currents, and provide significant advantages when identifying structures in sparse, randomly-positioned trajectory data. We extend this analysis to the exceptionally sparse sea ice domain by developing a temporal analysis of stretching exponent time series. Large TSE values correspond to proximity to hyperbolic LCS, which are regions of significant attraction, repulsion, and shear. We can then use single buoys to find spatial and temporal domains when sea ice is behaving like it is near an underlying hyperbolic LCS.

There are limited frame-indifferent Lagrangian alternatives for coherent structure identification with sparse buoy data to compare to. One notable exception is single-point (squared) relative dispersion (Haller and Yuan, 2000), also known as two-particle dispersion, which monitors the separation between a pair of particles. Similar variations have been applied in various contexts by Rampal et al. (2009), and Lukovich et al. (2014, 2015, 2017). These approaches are not evaluated here as one cannot generate both spatial maps and time series in a manner comparable to TSEs or (1-3). This is because relative dispersion relies on initially nearby buoys whose motions decorrelate at longer time scales. Relative dispersion diagnostics may show spatially coherent structures for a short window of time, but as the initially close buoys become decorrelated, one must reselect new buoy pairs, thus preventing a single meaningful time-series of local conditions as is possible with TSEs or (1-3). In this way, methods based on relative dispersion are more often used for understanding scaling relationships of sea ice motion.

In the present research, we show that TSEs localize regions and periods of significant stretching and compression, some of which is not evident with conventional buoy metrics. We independently verify the TSE stretching and compression features with two approaches. While TSEs are not a fracture-specific diagnostic, concurrent lead formation in remotely-sensed data provides a frame-indifferent confirmation of material sea ice response. We also compare our event detections with synoptic storm analysis from a high-resolution sea deformation experiment. This approach provides coupling between TSEs and the passage of storms previously verified to have significant sea ice influence. We discuss several logistical, and computational advantages with the single-buoy diagnostics, as well as future insights possible with TSE applications. We also compare our quasi-objective Lagrangian ice deformation analysis with standard Eulerian array-based rate-of-strain metrics.

## 2 Analytical Methods

For a continuously differentiable velocity field $\mathbf{v}(\mathbf{x}, t)$, a particle's trajectory $\boldsymbol{x}(t)$ is governed by the ordinary differential equation $\dot{\boldsymbol{x}}(t) = \mathbf{v}(\boldsymbol{x}(t), t)$, where $t$ represents time. Consider a material curve $\boldsymbol{\gamma}(t; s) \subset U \subset \mathbb{R}^2$, parameterized by the scalar parameter $s \in \mathbb{R}$ at time $t$ that has evolved from an initial curve $\boldsymbol{\gamma}(t_0; s)$. One can quantify the stretching of vectors tangent to this material curve using the equation of variations. In a steady flow (with no time dependence), particle trajectories are themselves material curves, and the Lagrangian velocity vector $\boldsymbol{v}(t)$ evolves as

$$\dot{\boldsymbol{v}}(t) = \nabla\mathbf{v}(\boldsymbol{x}(t))\boldsymbol{v}. \tag{5}$$

Haller et al. (2021) showed that the average material stretching over the time interval $t \in [t_0, t_N]$ of the Lagrangian velocity vector, $\boldsymbol{v}_0 = \mathbf{v}(\boldsymbol{x}_0)$, can be written as

$$\lambda_{t_0}^{t_N}(\boldsymbol{x}_0, \boldsymbol{v}_0) = \frac{1}{t_N - t_0} \int\limits_{t_0}^{t_N} \frac{d}{dt} \log \frac{|\mathbf{v}(\boldsymbol{x}(t))|}{|\boldsymbol{v}_0|} dt = \frac{1}{t_N - t_0} \log \frac{|\mathbf{v}(\boldsymbol{x}(t_N))|}{|\boldsymbol{v}_0|}. \tag{6}$$

For our situation, equation (6) gives an objective measure of sea ice stretching from only a single buoy velocity $\mathbf{v}(\boldsymbol{x}(t))$. This degree of stretching also only depends on initial and final conditions. That is, interim cycles of stretching and compression can cancel out. In sea ice, cycles of stretching and compression may lead to significant ridging and fracturing that one would not see in a standard fluid flow. To quantify the cumulative impact of repeated stretching, relaxation, or compression over a given time window, we can also utilize the averaged hyperbolicity strength with a strictly positive integrand,

$$\bar{\lambda}_{t_0}^{t_N}(\boldsymbol{x}_0, \boldsymbol{v}_0) = \frac{1}{t_N - t_0} \int\limits_{t_0}^{t_N} \left| \frac{d}{dt} \log \frac{|\mathbf{v}(\boldsymbol{x}(t))|}{|\boldsymbol{v}_0|} \right| dt. \tag{7}$$

Equation 7 adds up all hyperbolic (stretching and compression) action, giving a measure of cumulative changes, but does not differentiate stretching or compression. For slowly-varying (steady) flows, $\lambda_{t_0}^{t_N}$ approximates (equals) the finite-time Lyapunov exponent associated with the initial vector $\dot{\boldsymbol{x}}_0$ at the initial position $\boldsymbol{x}_0$, an objective Lagrangian measure of deformation over time (Ott and Yorke, 2008; Haller et al., 2021).

Remark 1: In an unsteady flow, the right hand side of Eq (5) would also include a $\delta_t\mathbf{v}(\boldsymbol{x}(t), t)$ term. For unsteady flows, the magnitude of the approximation error of tangential stretching in Eq (6) strongly depends on the slowly-varying nature of the flow, such as when Eulerian acceleration is much smaller than Lagrangian acceleration. The pointwise slowly-varying nature of sea ice velocities are assessed in Appendix A3 using Polar Pathfinder Daily 25 km EASE-Grid Sea Ice Motion Vectors (Tschudi et al., 2018), similar to the analysis conducted for ocean currents by Haller et al. (2022). Our assessment of the slowly-varying assumption does not depend on a specific integration time as it is a point by point comparison of Eulerian and Lagrangian acceleration. It does, however, depend on the temporal and spatial resolution of the underlying Pathfinder velocity field. A finer resolution velocity product could reveal sharper velocity gradients at smaller scales, such as across fractures, and further support the slowly-varying hypothesis. A shorter sampling period may also reveal stronger temporal gradients surrounding fracture events. However, this cannot be investigated until higher resolution and artifact-free velocity fields become available, and is a topic of future research.

Remark 2: Tangential stretching is rigorously defined when the trajectory is traveling in a continuous, incompressible medium. While it is true that sea ice velocity fields are not continuously differentiable, we can assume they are piecewise continuously differentiable (between fractures), and thus trajectory stretching exponents are well-defined in large continuously

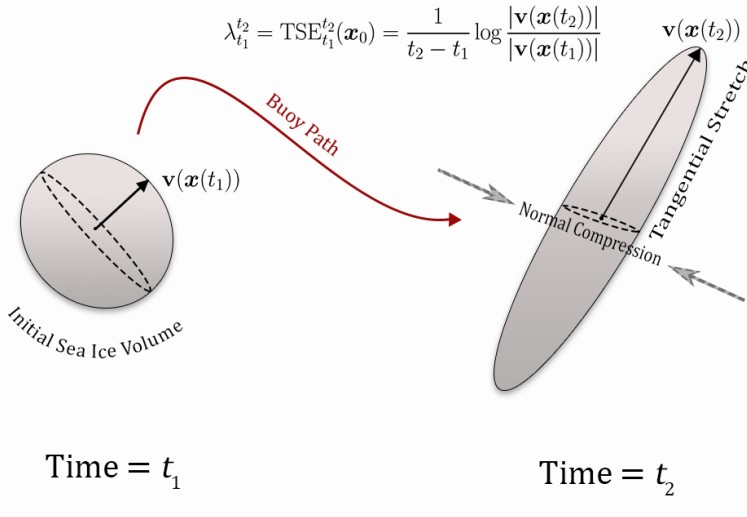

$$\lambda_{t_1}^{t_2} = \text{TSE}_{t_1}^{t_2}(\boldsymbol{x}_0) = \frac{1}{t_2 - t_1} \log \frac{|\mathbf{v}(\boldsymbol{x}(t_2))|}{|\mathbf{v}(\boldsymbol{x}(t_1))|}$$

**Figure 2.** Example evolution of a small volume of sea ice surrounding a buoy in a steady flow. From time $t_1$ to $t_2$, the buoy velocity vector $\boldsymbol{v}(t) = \mathbf{v}(\boldsymbol{x}(t))$ increases leading to compression in the plane of sea ice orthogonal to the buoy trajectory, and positive TSE.

deforming ice covers. In a steady incompressible two-dimensional flow, a negative stretching exponent, $\lambda_{t_0}^{t_N}$, equals the growth exponent of a vector normal to the material line formed by the trajectory $\boldsymbol{x}(t)$ (Figure 2) (Haller et al., 2021). For unsteady compressible flows, such as sea ice, $\lambda_{t_0}^{t_N}$ and $\bar{\lambda}_{t_0}^{t_N}$ are proxies for the material change in a plane normal to the trajectory, and may indicate stretching, ridging, or fracture. Negative $\lambda_{t_0}^{t_N}$ values indicate compressive processes whereas positive values

165  indicate stretching along the trajectory (Figure 2). The degree to which this differs from the growth exponent of the plane orthogonal to the trajectory depends on slowly-varying and compressibility conditions, as well as the rheology of the ice. These considerations parallel those for standard rate-of-strain metrics when dealing with a fractured ice cover, varying rheology, or regions with sea ice concentration less than 100%. For example, one is no longer solely quantifying the divergence of sea ice material when the ice is fractured and leads are present, rather the divergence of the mixed water-ice continuum. With these

170  considerations in mind, we focus primarily on mid-winter and early spring ice dynamics to minimize extensive fracturing of the ice cover, but briefly discuss dynamics during ice disintegration in Section 4.2. We further suggest due consideration when calculating stretching from ice buoys as ice concentration decreases and their trajectories become more representative of ocean dynamics than ice cover dynamics.

For real discrete observational trajectory data with an initial position $\boldsymbol{x}_0 = \boldsymbol{x}(0)$, we approximate the integrals in eqs (6) and

175  (7) and define the trajectory stretching exponents (TSEs) used for the remainder of this research:

$$\text{TSE}_{t_0}^{t_N}(\boldsymbol{x}_0) = \frac{1}{t_N - t_0} \log \frac{|\dot{\boldsymbol{x}}(t_N)|}{|\dot{\boldsymbol{x}}(t_0)|} \tag{8}$$

$$\overline{\mathrm{TSE}}_{t_0}^{t_N}(\boldsymbol{x}_0) = \frac{1}{t_N - t_0} \sum_{i=0}^{N-1} \left| \log \frac{|\dot{\boldsymbol{x}}(t_{i+1})|}{|\dot{\boldsymbol{x}}(t_i)|} \right|, \tag{9}$$

As with $\lambda$, TSE is positive for stretching, and negative for compression along a trajectory. $\overline{\mathrm{TSE}}$ does not allow for this cancellation as the summand is strictly positive, and gives a measure of cumulative changes, but does not differentiate stretching or compression, as with $\overline{\lambda}$. TSE and $\overline{\mathrm{TSE}}$ have been shown to accurately separate dynamically distinct regions (eddies and fronts) in sparsely-sampled open ocean flows (Haller et al., 2021), but have not yet been studied in a sea ice context.

In contrast to eqs. (2-3), TSEs do not differentiate between contributions from divergence or shear to the stretching of a material. Rather, they quantify stretching in the direction of vectors tangent to the buoy trajectory. We discuss the mathematical relationship between (2-3), the closely-related Lagrangian averaged divergence (Szanyi et al., 2016a), and TSEs in Appendix 1. TSEs are calculated using only buoy speed and do not require projection to orthogonal velocity components as in Green's theorem approximations from arrays. Speed can be easily calculated using geodesics between GPS locations, which prevents any inconsistencies of results due to map projections. Furthermore, TSEs are parameter-free with integration time being the only user-chosen value. The choice of integration time defines the average stretching extent, but does not prevent identification of events at timescales shorter or longer than that. See Section 4.1 for one such example of hourly timescale events being identified by a multi-day integration window.

The sampling period may influence the estimated speed of each buoy (Lei et al., 2021) and should be mentioned when performing analysis with TSEs. Once raw data has been collected, the user may then resample the data at a higher or lower resolution. Changing this sampling period is akin to changing the resolution of a Riemann sum that is approximating an integral. That is precisely what eqs. (8 & 9) are with respect to eqs. (6 & 7). The choice of interpolation method would have analogous effects.

While Eulerian strain-based metrics (2-3) provide an imperfect comparison to a Lagrangian Coherent Structure-based approach to ice deformation, they are commonly used in sea ice dynamics research, and generate long time series when polygons are suitably chosen (e.g., Hutchings et al., 2012; Itkin et al., 2017). TSEs are Lagrangian diagnostics and are calculated over a forward-looking window from a starting time, $t_0$ (eqs. 8 & 9). By incrementally increasing $t_0$, we can also generate time series of TSE and $\overline{\mathrm{TSE}}$ for each step in a buoy trajectory. In this way, large TSE and $\overline{\mathrm{TSE}}$ time series values should slightly precede significant storms, sea ice stretching, or breakup events as the summation occurs from $t_0$ forward in time. This also means that TSE and $\overline{\mathrm{TSE}}$ series will be slightly out of phase with concurrent Eulerian polygon-based divergence, shear, and deformation series. Indeed, we find this phase difference is on the order of the length of the integration window when maximizing the cross-correlation between TSEs and div and $D$ (Eq. 1-4, not pictured). We refer the reader to Appendix A1 for a thorough numerical comparison of Eulerian and Lagrangian diagnostics in a simple geophysical model.

Results here focus on the ability of TSEs to accurately characterize spatially and temporally localized coherent stretching during short-lived mid-winter storms. This choice is motivated by the availability of recent high-resolution experiments and local deformation being an open problem in sea-ice dynamics (Oikkonen et al., 2017). As with other Lagrangian methods, the integration time is user-defined and reflects the duration over which we are seeking significant and coherent stretching. For

example, previous LCS analysis using 6-month FTLE fields identified three distinct dynamic regions at pan-Arctic scales: the Fram strait, Beaufort gyre, and Northwind Ridge (Szanyi et al., 2016b). Similarly, TSEs are applicable to both short or long integration times $(t_N - t_0)$, and reveal time-averaged dynamics at those specified scales. We refer the reader to the Appendix A1 for a comparison of TSEs when varying integration times.

The choice of TSE integration window is only limited by the accuracy of the GPS trajectory data. Shorter integration times likely affected more significantly by small discrepancies in GPS locations. The instantaneous limit of the TSEs are also well-defined and provide approximations of hyperbolic (or parabolic) objective Eulerian coherent structures, which are the instantaneous limits of hyperbolic (or parabolic) LCSs (Serra and Haller, 2016; Nolan et al., 2020).

## 3   Datasets

Our results verify that TSEs can identify significant ice dynamics events using three different data sources at different spatial scales: synoptic storm analysis from a targeted experiment, linear kinematic feature (LKF) formation in subsequent SAR frames on the order of tens of kilometers, and sea ice brightness temperature measurements that show landfast and free-ice fractures at scales of hundreds of kilometers. We exhibit the utility of TSE and $\overline{\text{TSE}}$ spatio-temporal analysis for two experimental sea ice buoy data sets, the N-ICE2015 (Itkin et al., 2015, 2017) and MOSAiC expeditions (Krumpen et al., 2020), and data from the International Arctic Buoy Program (IABP) (International Arctic Buoy Programme, 2022).

### 3.1   N-ICE2015

The six-month Norwegian young sea ice cruise (N-ICE2015) sought to understand the rapid shift to younger and thinner sea ice, and its effect on energy fluxes, sea ice dynamics, and ecosystems in the Arctic basin. Numerous in-depth studies have evaluated the thin ice and weather conditions during the N-ICE2015 experiment (e.g. Cohen et al., 2017; Granskog et al., 2017; Itkin et al., 2017; Graham et al., 2019a). Itkin et al. (2017) conducted a thorough investigation of the ice response to N-ICE2015 storms, as identified by Cohen et al. (2017), which provides a test bed of buoy data where atmosphere-ice relationships during brief synoptic events are well understood (i.e. Itkin et al., 2015).

The experiment involved atmospheric, biogeochemical, oceanographic, and sea-ice dynamics components, including two separate buoy deployments from January to mid-March, and late April to June, 2015 (Figure 3). For our analysis, we focus on 24 buoy trajectories (Figure 3) in two time windows previously examined by Itkin et al. (2017). GPS positions were primarily sampled at 1-hour intervals, though some sampled every three hours. Itkin et al. (2017) resampled all trajectories to a 1 hr$^{-1}$ sampling frequency using a linear interpolant, and we follow this convention for our N-ICE2015 analysis. After interpolation, subsequent buoy speeds that exceeded 5 km/day were removed and buoy positions were resampled using a linear interpolant.

Itkin et al. (2017) calculated divergence, shear, and deformation using a tessellation of buoy triads, removing some of the user-dependence, but potentially introducing triads with inappropriate geometry for the contour integral approach. In particular, the winter deployment of buoys for the N-ICE2015 campaign was hindered by logistical challenges as the researchers deployed buoys in the polar night by snow machines and on skis. This resulted in an initially quasi-linear buoy array geometry, with many

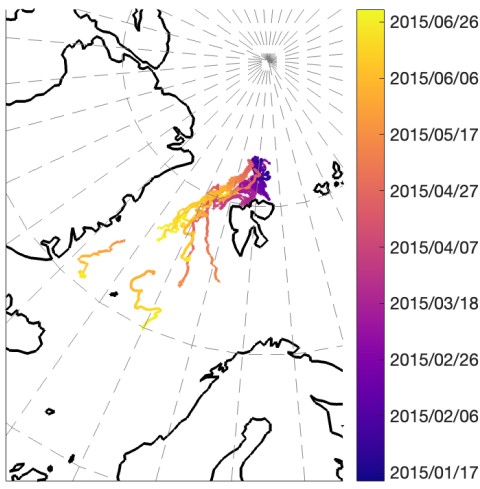

**Figure 3.** Trajectories of the N-ICE2015 buoys deployed north of Svalbard from January to June 2015 with positions colored by date.

of the triangles formed by buoy vertices having small angles ($< 15°$). Upon removing the most unreliable calculations, Itkin et al. (2017) successfully showed the impact of the Cohen et al. (2017) storms on shear, divergence and deformation signatures during the winter and spring deployments.

## 3.2 MOSAiC

The Multidisciplinary drifting Observatory for the Study of Arctic Climate (MOSAiC), was the largest multidisciplinary Arctic expedition to date, spanning the winter of 2019-2020 (Krumpen et al., 2020). The study centered around the research icebreaker Polarstern (Knust, 2017; Nicolaus et al., 2022; Rabe et al., 2022) which was moored to an ice floe for an entire year. MOSAiC provides an unprecedented look at winter ice dynamics with 213 unique buoy deployments and complementary atmosphere, ocean, ecology and biogeochemistry data.

We focus here on the paths of 101 buoys deployed within 40 km of the Polarstern. This public data set was documented by (Bliss et al., 2022). The half-hourly buoy track data was cleaned following (Hutchings et al., 2012). Triads were also handpicked from the MOSAiC buoys with data spanning October 2019 to June 2020, and is the focus of a forthcoming publication. The arrays were selected to maintain reasonable shapes (no small angles, area greater than 1km$^2$) from the beginning to the end of the time series and the public data was resampled to uniform 6-hourly intervals. Handpicking triads, however, does require user discretion. Buoy tracks were resampled to match the triad sampling rate. The arrays used are shown in Figure 4. A deeper comparison and refinement of geometrically suitable arrays in the MOSAiC data is a current topic of research. The method we use here is in line with previous work (Hutchings et al., 2011, 2012).

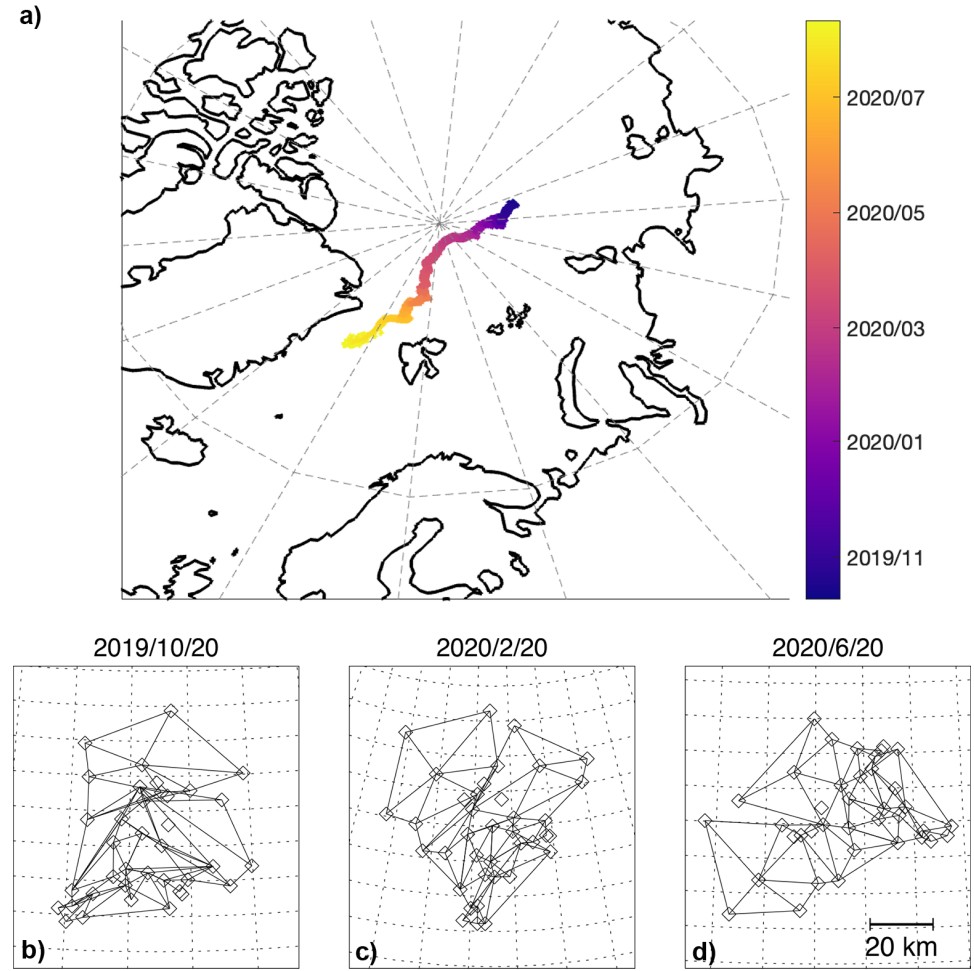

**Figure 4.** a) Trajectories of the MOSAiC buoys from October 2019 to July 2020 with positions colored by date. b-d) The MOSAiC buoys (diamonds) and triads (outlined with solid lines) as arranged for deformation calculation. The same array at three times is shown, with maps centered on (85.08N, 132.9E), (88.4N, 65.9E) and (81.9N, 8.3E) from left to right respectively. The scale of the maps is shown in the bottom right.

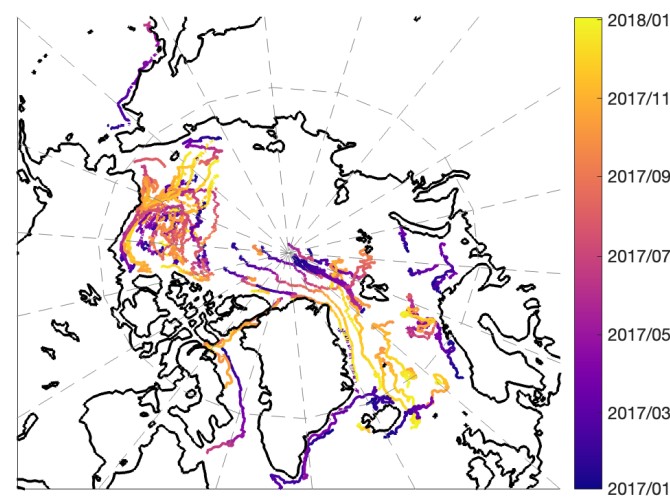

**Figure 5.** Trajectories of the IABP buoys from January 1, 2017 to January 1, 2018.

Though MOSAiC analysis is now just beginning, there is currently limited mid-winter buoy data available for the Arctic. MOSAiC and N-ICE2015 provide the best testbeds for evaluating the efficacy of TSEs to identify significant stretching events at high spatial resolution in coherent ice floes.

### 3.3   IABP

The international arctic buoy program (IABP, https://iabp.apl.uw.edu/data.html) is a network of drifting buoys deployed in the Arctic ocean by a collection of nations since the 1970's. In comparison with the previous two experiments, IABP buoy density is significantly lower, but the length of the IABP buoy record and its considerable spatial coverage allows for analysis of ice dynamics at much greater scales (see Figure 5). IABP data have been a useful ground-truth of sea ice motion for several decades, and continue to provide invaluable information for both model verification and understanding the complexities of ice dynamics (Rampal et al., 2009; Bouchat and Tremblay, 2020). IABP buoy tracks are available in both raw and cleaned formats. We focus our analysis on the publicly available 3-hourly L2 IABP buoy data. We downsampled the data a 6-hour sampling to maintain consistency with the MOSAiC data using a linear interpolant. We focus our analysis on trajectories during 2017 (Figure 5), with particular interest in the well-studied Beaufort sea.

## 4 Results

### 4.1 N-ICE2015

For our first evaluation of TSEs, we utilize previous storm analysis from the N-ICE2015 expedition. Cohen et al. (2017) studied winter storms during during this sea ice cruise, and found they typically occurred with a duration between 3 and 10 days. Using a buoy-array approach (Eq. 1-4), Itkin et al. (2017) found that sea ice deformation events could be roughly coupled with the passage of these storms. As we wish to quantify coherent ice deformation over a given time window, we choose a three-day integration period for stretching exponent calculations so as to capture the storm-scale Lagrangian stretching without averaging over too wide of a time window. In this way, TSE provides a before-and-after stretching measure, while $\overline{\text{TSE}}$ calculates the accumulation of trajectory stretching and contraction during the three-day window.

Figure 6 details TSEs for the winter and spring time windows analyzed by Itkin et al. (2017). The significant N-ICE2015 storms from Cohen et al. (2017), are shaded as pink regions and underly plots of TSE and $\overline{\text{TSE}}$ for all buoys in the two deployments. In winter, every storm of note was preceded by a local TSE extrema, including the short 10-hour storm on Jan 22, and the 5-hour storm on Feb 13. These two storms are well below the timescale of TSE integration we have chosen (three-days), but are still evident in the TSEs. This is because TSEs look at average behavior over a given time, not specifically events with a given duration.

Every coherent TSE peak above $1d^{-1}$ (marked by horizontal dashed line) was immediately followed by a significant storm in the TSE integration window, but local maxima do not always correspond to the passage of storms. This is likely due to oceanic influences not immediately related to the passage of storms. Itkin et al. (2017) calculated a significant divergence and shearing event for the buoy array on January 26, that did not correspond with the presence of a storm, but was instead related to a change in wind-direction. While TSE and $\overline{\text{TSE}}$ are both sensitive to accelerations caused by changes in wind direction, values in Fig. 6a-b do not reveal such a coherent stretching event, and indicate a possible advantage of using single-buoy stretching that is less reliant on buoy array geometry.

Storms during the spring deployment also coincided with significant stretching events, even for the sub-12-hour storms on May 16, May 30, and June 8. Similar to the analysis of Itkin et al. (2017), stretching time series for the spring deployment exhibited sub-daily oscillations, likely related to a more factured sea ice cover, tides, or inertial motions. There is also a large variance in the underlying buoy velocities, which is potentially an effect of using a 1-hour sampling window where small variations can lead to large velocity fluctuations. Due to these fluctuations, we find $\overline{\text{TSE}}$ is less informative in spring as an increase in small fluctuations has a positive cumulative effect during summation and reveals less about the storm-scale dynamics.

### 4.2 MOSAiC

Our second example utilizes MOSAiC buoy data from November 2019 to April 2020. We again choose a 3-day integration window in the absence of an alternative integration scale. Should an exhaustive storm study, such as that conducted for N-ICE2015, be performed, revisiting this analysis may be insightful. While the effect of integration scale does have some effect

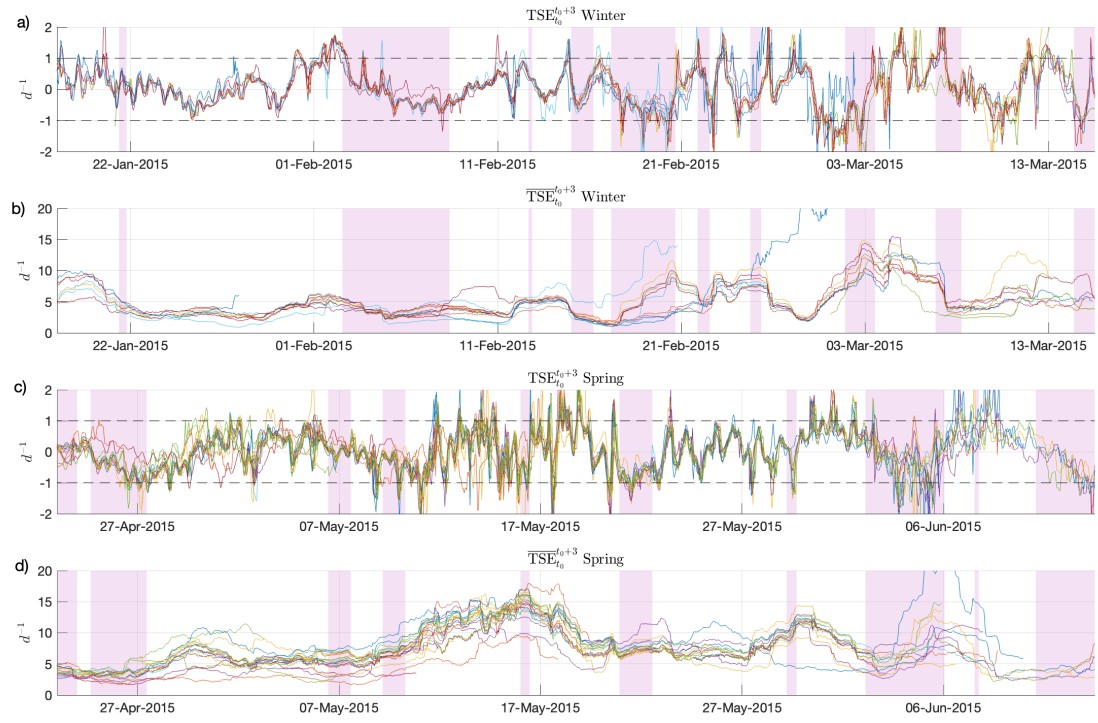

**Figure 6.** Comparison of TSE and $\overline{\text{TSE}}$ time series for 24 buoys during two stages of the N-ICE2015 experiment. Significant storm events for the winter deployment (a & b) and spring deployment (c& d) are shown as pink regions. A good correlation between strong stretching events and Arctic storms is shown: TSE extrema ($> 1d^{-1}$, marked by dashed lines) precede all storms and show no false positive sea ice stretching indicators.

on TSEs, identification of the underlying dynamical features is surprisingly robust (Appendix 1). In Figure 7a-d we compare time series of TSE (7a) and $\overline{\text{TSE}}$ (7c) with triad-based instantaneous approximations of divergence and deformation (eqs. 1-4). We plot all TSE time series available (101) and again find there is impressive coherence in stretching exhibited by the entire cluster. Local values of TSE and $\overline{\text{TSE}}$ also have the largest variance during peak stretching events. This allows us to analyze both the temporal localization of sea ice fracture by TSE, as well as potentially relate spatial differences of trajectory stretching to nearby LKFs. In Figure 7b and 7d, blue dots indicate the instantaneous value from each buoy triad, and the black lines show the mean over all triangles.

Qualitatively, prior to the January 31 event there is a reasonable agreement between TSEs and triad-based events, when accounting for the three-day integration window. For the entire period, one finds that all strong divergence and deformation events are accounted for as extrema ($> 1d^{-1}$) in the TSEs, though often with a different relative magnitude. In mid-November 2019, large TSE magnitudes generate a spike in $\overline{\text{TSE}}$ that is also evident as spikes in div and $D$. Local $\overline{\text{TSE}}$ maxima in mid

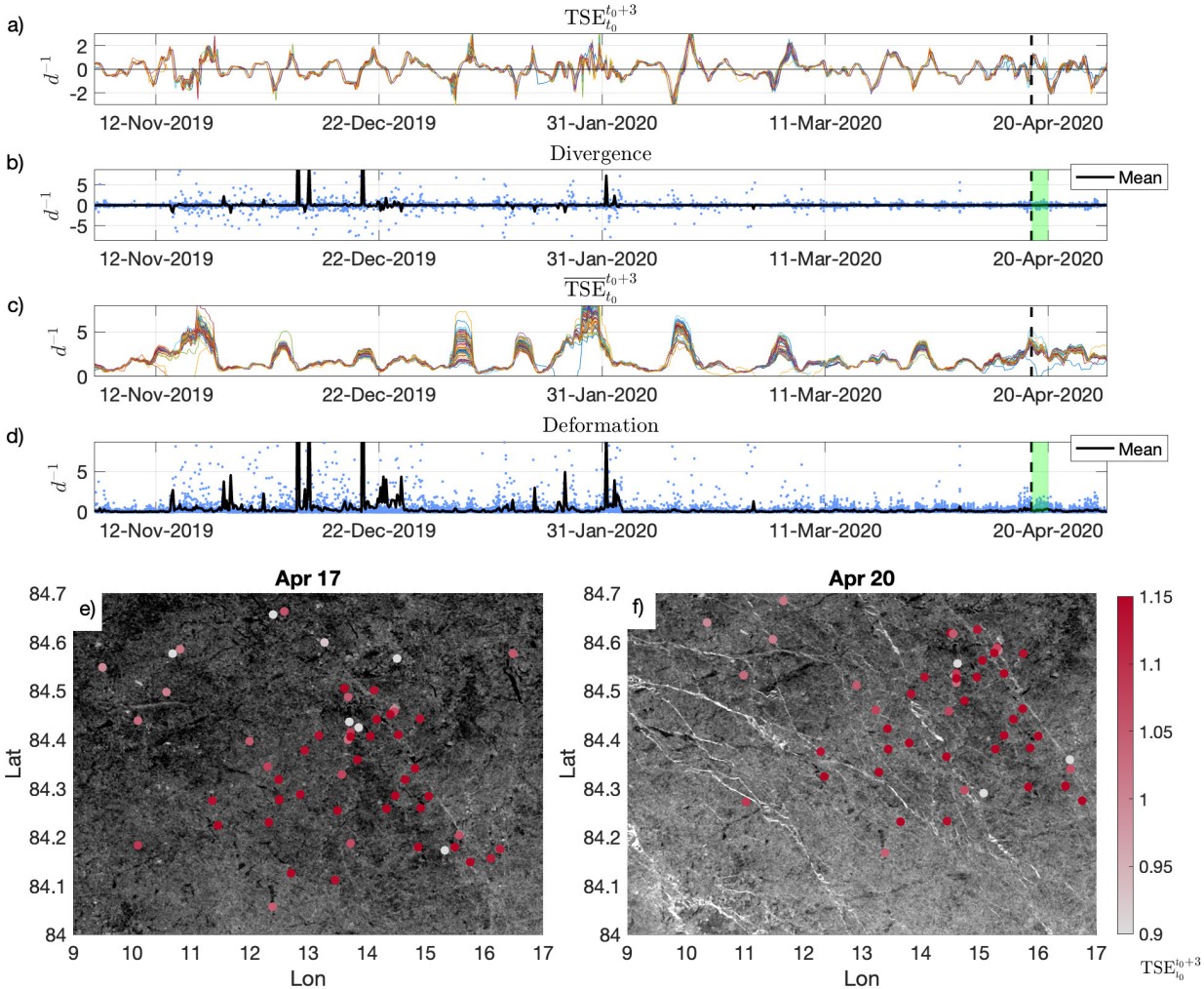

**Figure 7.** a-d) Time series comparison of trajectory stretching exponents and common buoy-array based approximations of divergence of and total deformation during the MOSAiC expedition. The vertical black dashed lines show a sea ice fracture event that was largely missed in the divergence and deformation. Panels e and f show a before and after comparison of HH Sentinel-1 frames surrounding the April 17 spike. Over this timespan, the creation of multiple LKFs in white can be seen, as well as the concurrent TSE values of nearby buoys.

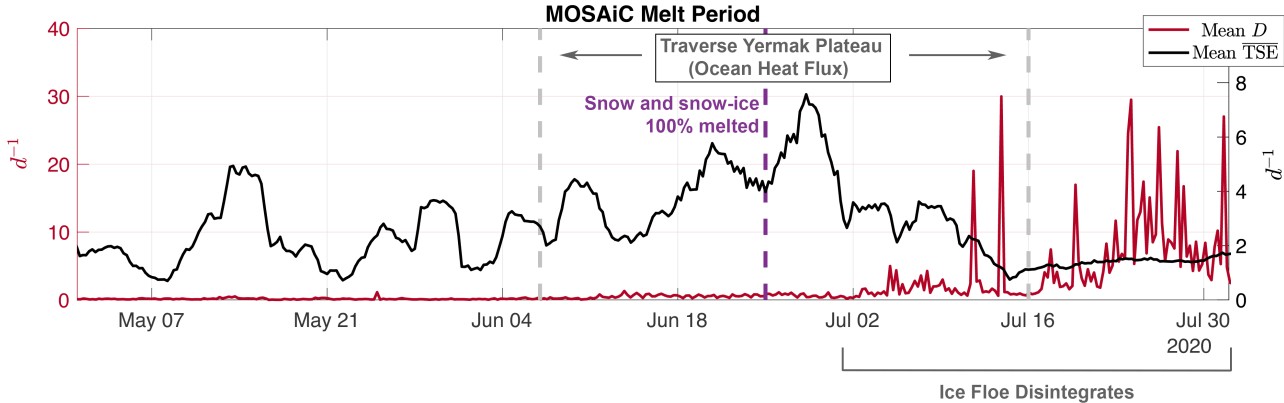

**Figure 8.** Average TSE and $D$ during the MOSAiC melt period. A gradual increase in trajectory stretching is found until the onset of ice floe disintegration. Peak stretching was found immediately after all snow and snow-ice was melted off the ice floe. Much larger $\overline{\text{TSE}}$ variance was found after a period of enhanced ocean heat flux as the ice floe crossed the Yermak plateau. At this stage, the ice concentration dropped below 60-85%, mean $\overline{\text{TSE}}$ values look flat from uncorrelated motions, and mean $D$ spikes are an order of magnitude larger than when the ice floe remained intact.

and late December also correlate well with spikes in div and $D$. Mean div provides a suggestion of whether divergence or convergence was more prevalent amongst the triads at a given time, but the significant scatter of both positive and negative values obscures such behavior at finer spatial scales. In contrast, there is much less ambiguity of the characteristic ice behavior for the entire cluster in the TSE series suggesting the cluster remained part of a largely coherent structure, that stretched and
compressed together, with variance between buoys appearing during dynamic events.

After January 31, several relatively large peaks in TSE appear as minor oscillations in div or $D$, or are not present. This indicates local dynamic stretching and potential fracture events may not be evident with the contour integral metrics. One such example is shown for the peak on April 17 highlighted by the vertical black dashed line. In div and $D$ plots, we also show the 3-day integration window in green. This example was chosen due to the concurrent availability of pre- and post-storm
SAR data during the period after the major January 31 event, as well as a relatively minor signature in div and $D$. The large stretching indicated by TSE on April 17, corresponds with multiple leads that formed throughout the domain of the MOASiC buoys. These fractures can be seen in the pair of HH polarization Sentinel-1 frames (Copernicus, 2020) (Fig. 7e-f), where white regions indicate new ice forming in leads or wind-driven waves (Grenfell et al., 1998). Points showing buoy locations have been colored by the April 17 TSE values. The spatial distribution of TSE values is difficult to interpret, and would require a
deeper analysis of the meteorological and ice conditions.

In the 3-day window following the Apr 17 TSE and $\overline{\text{TSE}}$ peak, the mean buoy divergence oscillated around zero (Figure 7b), with the magnitude staying below $0.1 d^{-1}$. This is approximately $1\%$ of peak values of mean divergence, suggesting a relatively insignificant period of divergence. This is in contrast to TSE and $\overline{\text{TSE}}$ on April 17 which sits at approximately $50\%$ of their total peak values, suggesting a relatively localized motion with a larger contribution to ice dynamics at the same time.

The contribution to total deformation from shear was slightly larger, with a peak mean deformation of $0.46d^{-1}$ at 17:00, April 17. The choice of triangles dictates which LKF are spanned during this dynamic event, and may explain why there was a weak deformation signature from the triads. As well, the lack of objectivity or trapezoid rule errors when using Green's theorem approximations with triads, as detailed in Appendix A2, may also contribute. In this scenario, the stretching and relaxation measured by TSE presents a clear correlation with material deformation of the ice and suggests TSEs may provide

ice behavior insight during times when Green's theorem methods are not possible, such as when there are too few buoys or they are their orientation is inappropriate, and when array-based approaches have underestimated dynamic behavior.

As sea ice melts in spring, ice concentration drops and the ice pack loses integrity, TSEs and rate-of-strain diagnostics are no longer strictly representing the deformation of the ice cover. In Figure 8, we look at mean $\overline{\text{TSE}}$ and mean $D$ during different stages of the melt period identified by (Lei et al., 2022). In late spring $\overline{\text{TSE}}$ values increase as the ice becomes more dynamic.

Once the ice floe enters warmer upper ocean water over the Yermak plateau, basal melting increases and $\overline{\text{TSE}}$ oscillations trend upwards. At the same time, oscillations in $D$ also begin to increase. By June 25, snow and snow-ice had completely melted from the sea ice cover, further enhancing the surface melt rates. After this, we obtain peak $\overline{\text{TSE}}$ values for the year, followed by a sharp drop. It is at the end of this sharp decline in stretching that Krumpen et al. (2021) note the ice floe begins to disintegrate. We hypothesize that $\overline{\text{TSE}}$ spikes precisely because this onset of disintegration enters the integration window.

At the timing of this drop in $\overline{\text{TSE}}$, and onset in disintegration, $D$ values also obtain large values never seen before in the flow. The sea ice concentration now drops to between 60% and 85%, depending on location of the buoy. From here on, buoy trajectories are no longer correlated and each buoy's $\overline{\text{TSE}}$ is measuring different dynamical flow features. At this stage, a mean $\overline{\text{TSE}}$ is no longer meaningful. However, future research may investigate spatially resolving single-trajectory flow features for heavily fractured ice floes, as has been done with analogous techniques in the open ocean (Encinas-Bartos et al., 2022).

## 4.3   IABP

In this section, we focus specifically on behavior in the Beaufort sea from October 2016 to October 2017. During this winter, thinner than usual sea ice may have caused the collapse of a typical high-pressure system over the Beaufort and anomalous surface winds (Moore et al., 2018). We again choose a 3-day integration window for our study to maintain comparability with the previous examples, though other timescales are equally applicable.

In Figure 9a, we show the three-day TSE for each buoy in the Beaufort during that period in grey. The maximum distance between buoys sampled here is approximately 30 times the lengthscale of the previous experiments. Due to the sparse sampling over a much larger spatial domain, there is much more variance around the mean TSE value, shown in black in Figure 9a. The absolute maximum of mean TSE during this period is indicated by a red dot, signifying a period of maximal stretching and deformation over a large domain. The following two peaks are also identified by red dots. These events correspond with three

dynamic shifts in Beaufort sea ice structure and behavior. The frame pairs in Figure 9b-d show before and after frames of AMSR2 Sea Ice Brightness Temperature (89V GHz) (Meier et al., 2018) for times corresponding to high TSE values. Buoy positions are again colored by TSE in the initial frame and colormaps are consistent across all frames. A full animation of this breakup can be found in the supplementary video.

The first event corresponds to stretching from March 26 to March 29, 2017. Previous mean TSE oscillations gradually
increased to the absolute maximum of mean TSE on March 26. Prior to any evidence of detachment of the mobile pack ice in
the Beaufort sea, TSE values were indicating an ongoing increase of stress and strain leading to the major fracturing in March
and April, 2017.

Between March 26th and 29th, an initial significant fracture formed when the mobile Beaufort pack ice separates from the
landfast ice on the northern coast of Alaska. The fracture extends to just off the coast of Banks Island (BI in Figure 9c) as the ice
in the Beaufort Gyre begins to freely rotate in an anticyclonic motion. All buoys in the free-drift region have large positive (red)
TSE values and create another local maximum in the mean time series, further supporting this relatively significant stretching
event in March and April when compared to times prior to and after these months. From April 3 to April 8, significant fracturing
throughout the region south of $76°$ N occurred, with another major fracture extending off the northwest corner of Banks Island.
The buoys in the region of this enhanced fracturing showed large positive TSE values, whereas buoys further north without
new fracturing showed low TSE values. The last pair of frames show the last major shift in ice dynamics from April 13 to
April 17. At this time, a major lead formed off of Prince Patrick Island (PI). This was the last stage of the changing dynamics,
after which all ice south of the PI lead exhibited a much freer anticyclonic rotation, decoupled with the surrounding ice. Again,
the largest positive TSE values were found in the spatial domain where significant ice breakup was occurring, with lower TSE
values, hundreds of kilometers away, delimiting the fracture domain to the North. After the April 17 fracturing, TSE values
remain relatively low compared to these distinct peaks, similar to the behavior of the MOSAiC buoys during the disintegration
of the ice floe in Section 4.2.

One specific benefit displayed in this example is the significant spatial extent of the large positive TSE values prior to each
fracture in the Beaufort gyre. Not only was the edge of the gyre identified in the gap between positive and negative TSE, but
positive TSE was also found 1000 kilometers from the Prince Patrick and Banks Island fractures on the western edge of the
gyre. This supports the ability of TSE to identify the Lagrangian coherent structures in the mobile pack ice as whole, not just
locally highlight a fracture.

## 5   Conclusions

We find that TSEs successfully identify significant local material deformation tangent to individual sea ice buoy trajectories.
Periods of strong local stretching are representative of changing ice dynamics in a neighborhood to 10-1000 km. That is, though
TSEs are local in nature, their values provide insight into much larger coherent sea ice structures, and distant fracture events.
This local-regional connection provides a valuable avenue for researching sea ice dynamics in sparsely sampled regions, and
understanding the changing rheology of sea ice. In contrast, conventional polygon-based approaches provide an estimate of the
spatial-average of divergence, shear, or deformation over an ice volume, with errors increasing as area decreases, number of
buoys decreases or the array becomes skewed.

Approaching sea ice dynamics through quasi-objective stretching, we were able to capture coherent deformation events
in concentrated and sparsely-sampled buoy experiments. Spatial and temporal signatures of stretching with TSE are well

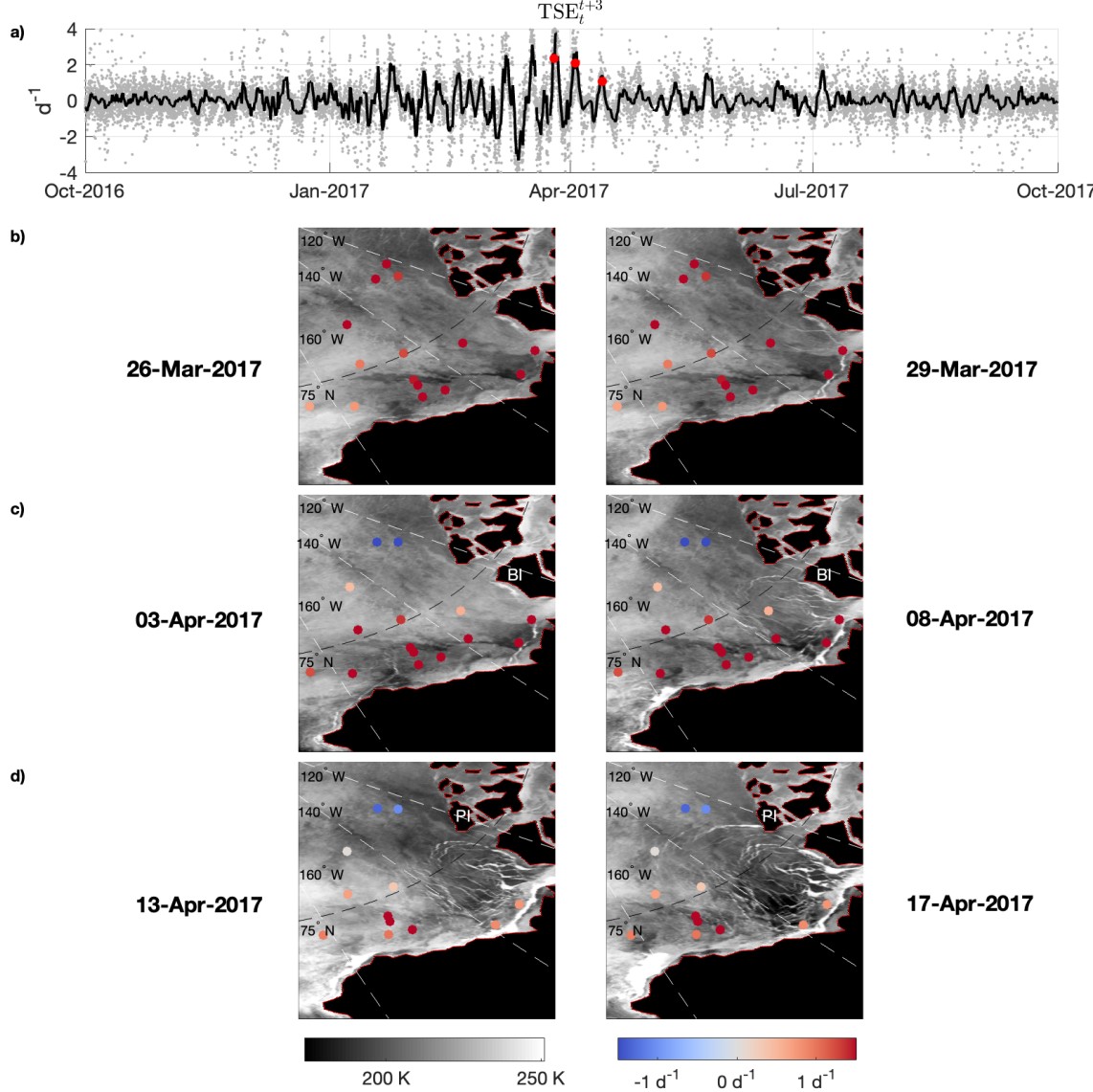

**Figure 9.** a) Time series of TSE for buoys in the Beaufort sea from October 2016-2017, with mean shown in black. Red points correspond to local maxima of TSE as the ice breaks into a state of free-drift. b-d) The three steps of the Beaufort gyre sea ice freeing itself from surrounding ice beginning its spring anticyclonic acceleration, corresponding with the three highlighted TSE spikes in a). Before and after brightness temperature images show the significant fractures at the northern coast of Alaska (b), Banks Island (c), and Prince Patrick Island (d).

correlated with formation of nearby leads and changing ice transport patterns, as well as with the concurrence of well studied sea-ice-impacting storms. For two high resolution mid-winter buoy deployments, we find that the TSE had greater sensitivity to sea ice deformation than the common polygonal approach, and potentially avoided a false positive identification.

Specifically, large TSE values coincided with major storms in the N-ICE2015 experiment, and did not identify a mysterious storm-free dynamic ice-event that was described by Itkin et al. (2017). For the first half of the MOSAiC experiment, we found a good qualitative agreement between polygon-based metrics and the TSEs, though this correlation was much weaker following a large midwinter storm. During this latter half, we verified TSEs were able to identify an influential stretching event that was much less evident with Green's theorem methods. Lastly, TSEs were able to spatially and temporally isolate major fracture
events during Beaufort sea ice breakup using IABP data at a much lower spatial resolution. A buildup of stress in the Beaufort sea was detailed by increasing local TSE maxima, until an initial separation formed at the land-fast ice during the seasonal TSE peak. During subsequent diminishing peaks the ice fractured and further detached from land-fast regions, creating a coherent mass of rotating ice that was delimited spatially with TSE. Increasing amplitude TSE events preceded spring breakup events in both high-resolution MOSAiC and sparsely-sampled IABP data. This suggests predictive abilities of TSE may possibly be
developed with further investigation.

The single-buoy quasi-objective trajectory stretching exponents (TSEs) identify dynamic sea ice events that are potentially significant in terms of understanding spatially and temporally varying sea ice deformation. As sea ice dynamics plays an important role in atmosphere-ice-ocean exchange processes, we find the further event-detection sensitivities possible with TSEs are a valuable complement to common, polygon-based divergence, shear, and deformation approximations. We find TSE
usage provides some distinct advantages and potential improvements for future research:

1) TSEs identify deformation events from a single buoy and are thus unaffected by buoy array geometry, or the number of buoys deployed. As buoys passively follow ice floes, we can still obtain insights into ice dynamics as arrays become heavily skewed and non-uniform. TSE maintains integrity in this situation, whereas arrays of buoys become aligned and unsuitable for strain estimates due to shearing in the ice pack. This expands the domain of potential analysis, and reduces logistical burdens
of Arctic and Antarctic expeditions as stretching dynamics can be studied with TSE from linear, randomly distributed, or organized arrays. This is particularly valuable for the Antarctic where there are significantly fewer buoys deployed than in the Arctic.

2) As TSEs are quasi-objective metrics in continua, their values provide a useful proxy for frame-indifferent (objective) measures of stretching, with a quantitative difference depending on the compressibility of the ice and slowly-varying nature of
the flow. In a heavily fractured ice context, the degree to which these single-trajectory metrics approximates along-trajectory ice stretching varies, but this work suggests positive correlations with remotely-sensed dynamic fracture and breakup events. Further numerical and observational experiments can help improve these correlations.

3) TSE values do not need to be separated or averaged based on length scales (e.g. Itkin et al., 2017), and showed great success in identifying fracture regions at a wide range of spatial scales.
4) TSE calculations are mathematically simple as TSE is calculated using only buoy speed and does not require projection to orthogonal velocity components. Speed can be calculated using geodesics between GPS locations, which prevents any

inconsistencies of results due to choice of map projection or projection distortions, or needing to perform differential calculus in an elliptic geometry. Furthermore, TSE is parameter-free with integration time being the only user-chosen value.

Calculating TSE from single buoy trajectories requires significantly less effort than SAR-based approaches, is not subject to
the same errors (e.g. Bouchat and Tremblay, 2020), and also supplements data in the pole hole during winter (e.g. Krumpen et al., 2021). If we can further verify the slowly-varying nature of sea ice at sub-diurnal time scales, TSE may also fill in temporal gaps due to the sparse monitoring of satellites. Calculating TSE fields from more general feature trajectories, such as from X-Band and HF radar or SAR datasets, would be a straight-forward application of Eq (8) & (9) and can also enrich the analysis of sea ice dynamics in existing datasets.

The ability of TSEs to overcome some of the short-comings of other buoy approaches may also provide an additional source of deformation information, such as is necessary to constrain and improve sea-ice models (Bouchat and Tremblay, 2020). To obtain rate-of-strain invariants for sea ice deformation, it is still necessary to use a high-density buoy array. Such an array also reveals gradients of trajectory stretching and further enhances precise stretching localization with TSEs. A modeling study of compressibility, slowly-varying impacts, and sampling frequency would be useful for TSE applications in model
development, as would a further comparison with divergence, deformation, and shear to aid the broader community in the physical interpretation of TSE signals. This would help interpret TSEs as complementary sources of stretching information when buoy arrays are aligned linearly due to strong shear, have edges that span large distances encompassing multiple fractures or coherent structures, or have areas below tolerable error thresholds. In remote polar regions where data is still difficult to obtain, but the changing climate has an outsized impact, TSEs provide new physics-based insights into ice dynamics while
only requiring single GPS tracks.

Our findings suggest that spatio-temporal analysis of local and regional sea ice dynamics should be revisited with a comparison between single-trajectory and spatial velocity gradients. Our comparisons suggest potentially significant dynamic events in terms of understanding atmosphere-ice-ocean exchange processes may go undetected using conventional polygon-based methods, or in the case of N-ICE data, may be inaccurately detected. This advantage stems from the TSEs' ability to identify
stretching in a small neighborhood of individual buoys and connect local changes with broader ice behavior. A deeper understanding of TSEs connection to more broadly used ice dynamics metrics will help researchers understand how TSE can inform ice responses to various forcings.

**Appendix A: Introduction**

The first section of the Appendix develops the rigorous connection of trajectory stretching exponents to more common de-
formation diagnostics through the rate-of-strain tensor and Cauchy-Green strain tensor. The second section of the Appendix provides a simple analytic example of buoy motion to illuminate Green's theorem estimation errors for a buoy-array diagnostic. This example shows how the Green's theorem approximation fails to provide self-consistent divergence quantification (in a divergence-free flow), depending on equilateral triad orientation. The last section of the Appendix provides an assessment of the assumption that sea ice is a slowly varying flow to satisfy criteria for the quasi-objectivity of TSEs.

## A1 Relating TSE to Shear and Divergence

### A1.1 Background

From the gradient of a two-dimensional velocity field $\mathbf{v}(\boldsymbol{x}, t)$,

$$\nabla \mathbf{v} = \begin{pmatrix} \frac{\partial u}{\partial x} & \frac{\partial u}{\partial y} \\ \frac{\partial v}{\partial x} & \frac{\partial v}{\partial y} \end{pmatrix}$$

we can define the rate of strain tensor

$$\mathrm{S}(\boldsymbol{x}, t) = \frac{1}{2}(\nabla \mathbf{v} + \nabla \mathbf{v}^T). \tag{A1}$$

We can express the common Eulerian sea ice deformation metrics, divergence and shear, as functions of the eigenvalues (principal stresses) of (1):

$$div = \mathrm{tr}(\mathrm{S}) = \lambda_1 + \lambda_2 \tag{A2}$$

$$shr = \sqrt{\mathrm{tr}(\mathrm{S})^2 - 4\det(S)} = \sqrt{(\lambda_1 - \lambda_2)^2 - 2\lambda_1\lambda_2} \tag{A3}$$

$$D = \sqrt{shr^2 + div^2} \tag{A4}$$

To quantify the deformation of a material over time, one considers the flow map

$$F_{t_0}^t(\boldsymbol{x}_0) = \boldsymbol{x}(t; t_0, \boldsymbol{x}_0) \tag{A5}$$

that takes an initial position $\boldsymbol{x}_0$ to its current position $\boldsymbol{x}(t; t_0, \boldsymbol{x}_0)$ following the trajectory defined by the different equation

$$\dot{\boldsymbol{x}} = \mathbf{v}(\boldsymbol{x}, t).$$

From the gradient of the flow map, we define the right Cauchy-Green strain tensor as

$$C(\boldsymbol{x}_0) = [\nabla F_{t_0}^t(\boldsymbol{x}_0)]^T \nabla F_{t_0}^t(\boldsymbol{x}_0) \tag{A6}$$

Invariants of $C$ are commonly used to study the deformation of material in continuum mechanics (Truesdell and Noll, 2004). In an incompressible two-dimensional flow, the rate of length change for an infinitesimal material element vector $\boldsymbol{l}$ based at the point $\boldsymbol{x}$ is

$$\frac{1}{2}\frac{d}{dt}|\boldsymbol{l}|^2 = \langle \boldsymbol{l}, \mathrm{S}(\boldsymbol{x}, t)\boldsymbol{l} \rangle. \tag{A7}$$

Likewise, consider a curve $\gamma$ parameterized by the dummy-time variable $s$ that is evolving under the flow map, $F_{t_0}^t(\boldsymbol{x}_0)$. This could represent the evolution of a line of dye in a fluid, or a physical transect along sea ice. A vector $\boldsymbol{\xi}_0$ tangent to $\gamma$ at $\boldsymbol{x}_0$ evolves as $\boldsymbol{\xi}_t = \nabla F_{t_0}^t \boldsymbol{\xi}_0$. It then follows that

$$|\boldsymbol{\xi}_t|^2 = \langle \nabla F_{t_0}^t(\boldsymbol{x}_0)\boldsymbol{\xi}_0, \nabla F_{t_0}^t(\boldsymbol{x}_0)\boldsymbol{\xi}_0 \rangle = \langle \boldsymbol{\xi}_0, C_{t_0}^t \boldsymbol{\xi}_0 \rangle. \tag{A8}$$

In the case that $\boldsymbol{l}$ or $\boldsymbol{\xi}_0$ are eigenvectors of their respective tensors, (A7) and (A8) equal the respective eigenvalues $\lambda_i$, which represent the stretching of an infinitesimally small sphere into an ellipse along its principal axes Haller (2015). When $\boldsymbol{\xi}_0$ is the eigenvector associated with the the larger Cauchy-Green eigenvalue, $\lambda_2$, one easily obtains the widely-used finite-time Lyapunov exponent,

$$\text{FTLE} = \frac{1}{t_1 - t_0} \log(\lambda_2(C)).$$

Otherwise, one can define the averaged stretching exponent

$$\lambda_{t_0}^t = \frac{1}{t - t_0} \log \frac{|\boldsymbol{\xi}_t|}{|\boldsymbol{\xi}_0|}, \tag{A9}$$

also known as the finite-time Lyapunov exponent associated with the initial vector $\boldsymbol{\xi}_0$. Due to the exponential growth of fluid particle separation and chaotic nature of fluid flows, one often utilizes stretching exponents in this context instead of eigenvalues of deformation tensors, though they are clearly closely related.

Lagrangian averaged versions of shear and divergence can also be computed as

$$LS_{t_0}^t = \frac{1}{t - t_0} \int_{t_0}^{t} shr(\boldsymbol{x}(s))ds, \qquad LAD_{t_0}^t = \frac{1}{t - t_0} \int_{t_0}^{t} div(\boldsymbol{x}(s))ds, \tag{A10}$$

respectively. LAD was previously used by Szanyi et al. (2016a) to differentiate between different contribution to FTLE at pan-Arctic scales.

To calculate the rate-of-strain tensor, once must calculate spatial derivatives of the velocity field. Similarly, the right Cauchy-Green strain tensor requires the spatially and temporally resolved velocity field information to accurately calculate the gradient of the flow map. This sort of spatially and temporally resolved velocity data is unavailable from sparse buoy trajectories. TSEs are designed to complement deformation metrics in sparse data settings.

Consider a trajectory $\boldsymbol{x}(t)$. One can calculate the Lagrangian velocity vector along the trajectory

$$\boldsymbol{v}(t) = \mathbf{v}(\boldsymbol{x}(t)) = \dot{\boldsymbol{x}}(t),$$

but we do not have any information about nearby velocities. Haller et al. (2021) show that, in a reference frame where the velocity field is steady (does not change with time), $\boldsymbol{x}(t)$ is a material curve, and $\boldsymbol{v}(t)$ evolves as a tangent vector to the

trajectory. Thus the stretching of $\boldsymbol{v}(t)$ is a measure of material stretching along the trajectory and we can closely approximate the averaged stretching exponent in that frame without the need of spatial derivatives. That is, in a steady flow (no time

dependence), TSE measures the correct stretching of Lagrangian velocity vectors as they materially evolve in that reference frame, as determined by the gradient of the flow map (A8).

The instantaneous limit of TSE corresponds with the analogous Eulerian rate of length change (A7) for the choice of vector $\boldsymbol{l} = \boldsymbol{v}(t_0)$. In the following examples we highlight the ability of TSE and $\overline{\text{TSE}}$ to delineate distinct coherent flow features without the use of spatial derivatives and compare their behavior with other diagnostics. These models are idealized flows

without many of the complication from sea ice motion. That is, they represent steady flow incompressible flow fields of a continuous medium and are thus ideal for comparing these diagnostics.

### A1.2   Examples

For our first example, we compare Eulerian and Lagrangian diagnostics for the widely studied model of geophysical fluid flow (see, e.g., Rypina et al., 2007), the steady Bickley flow. The stream function of the flow is given by

$$\psi(x_1, x_2) = cx_2 - U_0 L \tanh(x_2/L) + AU_0 L \text{sech}^2(x_2/L) \cos kx_1 \qquad \text{(A11)}$$

where we use the geophysically based parameters $U_0 = 62.66\text{e}{-}6$, $L = 1.77$, $c = U_0/2$, $A = 0.1$, $r_0 = 6.371$, $k = 6/r_0$. As the velocity field is derived from the stream function, the velocity field is divergence-free. The flow consists of a central westward jet, two eastward jets above and below, and clockwise vortices. Figure A1 shows streamlines and velocity vectors for this flow, as well as divergence, shear, and vorticity fields calculated on a $500{\times}150$ grid. Note that the $\text{div} = 0$ by design, and that the

vortex cores align when comparing streamlines and vorticity contours. Shear (Fig A1c), shows a shear-free core, and high shear regions adjacent to the jet and vorticies, though contours of shear do not agree with the topology suggested by the streamlines or vorticity. This indicates that shear is not a good indicator of the edges of all coherent structures.

Figure A2 details TSE and $\overline{\text{TSE}}$ fields for a $500{\times}150$ grid of initial conditions, calculated from a trajectory integration time of 90 days. We also calculate LS (Fig. A2c) for each trajectory $\boldsymbol{x}(s)$. For brevity we omit LAD as the flow is divergence free.

Contours of TSE or $\overline{\text{TSE}}$ organize particles with similar trajectory stretching and reveal coherent structures of the flow. Note that both TSE or $\overline{\text{TSE}}$ can identify the edges of the central jet and eddies as shown in the streamlines of this steady flow. $\overline{\text{TSE}}$ also reveals where cumulative trajectory-tangent stretching is greatest, near the edges of the eddies. TSE provides intricate stretching and compression information surrounding the vortex cores and hyperbolic saddles, suggesting a complex stretch-relax cycle for material in the flow. $LS$ also does a much better job than Eulerian shear to extract the boundaries of each

feature, and also highlights the stronger shear near the edges of the central jet.

These underlying structures, and their subsequent influence on sea ice deformation is exactly the kind of information that TSEs provide for sea ice dynamics analysis. As can be seen, mathematically there is limited relationship between the Eulerian metrics (eqs. A2-A4), and the stretching exponents (A9), particularly because $\boldsymbol{\xi}$ is not necessarily aligned with the eigenvectors

of S. Qualitatively, one can see there is no relation between div and TSEs for this simple example, and very limited similarity with shear, except when evaluating a Lagrangian average ($LS$).

We also examine the influence of integration time on TSEs for this steady flow. In Figure A3, we increase the integration time of particle trajectories from 1 day to 30 days and plot TSE and $\overline{\text{TSE}}$ contours. The eddies, saddles between eddies, and central jet are evident for all integration times, but the clarity of boundaries increases as integration time increases. Note this improvement is not always possible with time-dependent flows as features may change. In more complex turbulent flows, such clarity at higher integration times can also be hindered by trajectories traveling near multiple uncorrelated coherent structures.

For our second example we use there is in fact no reliable relationship to be inferred between TSEs and shear with a simple divergent flow

$$v_1 = x_1 \tag{A12}$$

$$v_2 = x_2. \tag{A13}$$

It is easy to analytically calculate that shear and vorticity are constantly zero, and divergence is uniformly equal to 2. This is reflected in the contour plots of Figure A4. Stretching is also constant along trajectories in this flow due to the linear acceleration away from the origin. In this way, TSE and $\overline{\text{TSE}}$ reveal the same uniform deformation structures as the divergence field (Figure A5). As shear is everywhere zero, $LS$ is also zero, and reveals the same uniform structure as the TSEs, divergence, and $LAD$.

### A1.3 Conclusion

The previous mathematical exposition and examples show there is no simple relationship between TSE and $\overline{\text{TSE}}$ with $div$, $shr$, or vorticity. This is in part because TSE and $\overline{\text{TSE}}$ are Lagrangian metrics, they measure material deformation over time, whereas $div$, $shr$, and vorticity are Eulerian, relying only on instantaneous rates of change at one point in time. TSE and $\overline{\text{TSE}}$ do not approximate $div$, $shr$, or vorticity.

As such, TSE and $\overline{\text{TSE}}$ fields are most successfully used for identifying dynamic flow features, such as the edge of a gyre, or deformation across a jet. This stems from an integration of temporal changes in material stretching along a trajectory. As such, Lagrangian averaged shear also revealed more about the flow structures than Eulerian shear. Temporal analysis of TSE and $\overline{\text{TSE}}$ time series, as conducted in the research presented here, reveals changes in deformation immediately around $\boldsymbol{x}(t)$ as a buoy transitions from one flow feature to another. For an additional in-depth comparison with TSE and $\overline{\text{TSE}}$ and other Lagrangian metrics, one is referred to Haller et al. (2021).

### A2 Trapezoid rule errors for buoy arrays in a continuous flow

Consider the incompressible 2D flow

$$u(x,y) = -y^2, \qquad v(x,y) = x^2. \tag{A14}$$

One finds $\partial u/\partial x(x,y) = \partial v/\partial y(x,y) = 0$, and $\text{div}(x,y) \equiv 0$. Consider also the equilateral triangle formed by three buoys positioned on the unit circle, $(x_{b1}, y_{b1}) = (\cos(\frac{\pi}{2}), \sin(\frac{\pi}{2}))$, $(x_{b2}, y_{b2}) = (\cos(\frac{7\pi}{6}), \sin(\frac{7\pi}{6}))$, $(x_{b3}, y_{b3}) = (\cos(\frac{-\pi}{6}), \sin(\frac{-\pi}{6}))$.

We estimate the "Green's theorem divergence" for this triad as $\text{div}_G = -0.5$. Though this flow is entirely divergence free, the value of $\text{div}_G$ is sensitive to the triad's rotation on the unit circle. Figure S2a shows the range of $\text{div}_G$ calculated as the triad is rotated. Depending on how the vertices are oriented, this approach will suggest either a divergent or convergent flow, with the correct value only appearing for two specific orientations.

In contrast to errors that have been discussed and quantified before (e.g., Lindsay and Stern, 2003; Hutchings et al., 2012;

Dierking et al., 2020), this issue arises from the approximation of the partial derivative contour integrals by the trapezoid rule. For a generic real-valued twice differentiable function $f(x)$ defined on the interval $[a,b] \subset \mathbb{R}$, the difference between the true integral and trapezoid rule is bounded by (Atkinson, 1989):

$$E = -\frac{(b-a)^3}{12} f''(y) \qquad y \in [a,b]. \tag{A15}$$

That is, the potential error of each partial derivative scales with the cube of the distance between buoys and the second derivative

of the associated velocity component along that interval. If we increase the number of uniformly distributed buoys on the unit circle in this velocity field, we find the average divergence approximation inside the polygons quickly converges to the true value (Figure S2b). In fact, using a four sided polygon can significantly reduce the error for this simple flow, as an equal number of buoys can be positioned on either side of the axis dominant flow structure (Figure S2c). This improvement can be expected for other flows as well, though more complex flow structures would require a greater number of vertices to avoid

these same errors. Analogous errors are evident in Green's theorem approximations of shear and total deformation as well, reminding researchers to consider the number of buoys and methods used when quantifying sea ice dynamics.

### A3   Slowly-varying condition of sea ice

TSE is a quasi-objective metric, indicating it approximates an objective metric under a given condition. For the work presented here, this requires that the velocity field of interest be slowly varying. That is, $|\mathbf{v}_t(\boldsymbol{x}(t), t)| \ll |\mathbf{a}(t)|$, where $\mathbf{a}(t)$ is the La-

grangian velocity. In order to calculate the temporal derivative on the LHS, we rely on daily gridded sea ice velocity data from Tschudi et al. (2018) as a best estimate for the flow. While one can calculate $\mathbf{a}(t)$ from trajectory data, discrete calculations of $\mathbf{v}_t(\boldsymbol{x}(t), t)$ rely on repeat measurements of velocity at the same location. In this way, we require gridded velocity data to validate the slowly-varying assumption.

The use of Pathfinder sea ice displacement grids comes with its own errors, such as the spatial gradient artifacts previously

identified by Szanyi et al. (2016a). Furthermore, the low temporal resolution of this data smooths the short-time variability of surrounding instantaneous shear and fracture events. Prior to the development of a more robust gridded dataset of ice velocities, this remains the best available option.

The probability function in figure A7 shows that for the majority of the Arctic domain, the slowly varying condition is satisfied pointwise along sea ice trajectories. This probability is representative of mid-winter ice conditions, and may vary

in the summer months. As this is a pointwise ratio, and not an integrated comparison, the slowly-varying condition does not depend on a specific time window, the resolution of the underlying Pathfinder velocity field.

*Code availability.* MATLAB scripts for calculating trajectory stretching exponents can be found here: https://github.com/NikAksamit/TRA_TSE

*Data availability.* The ocean flow data used for Figs. 1 can be obtained from AVISO (http://www.aviso.oceanobs.com, last access: 19 November 2019). The MOSAiC drifter data (Bliss et al., 2021) is available here: https://arcticdata.io/catalog/view/doi:10.18739/A2Q52FD8S, the N-ICE2015 data (Itkin et al., 2015) is available here: https://doi.org/10.21334/npolar.2015.6ed9a8ca, the IABP data (International Arctic Buoy Programme, 2022) is available here: http://iabp.apl.uw.edu/data.html, the AMSR data (Meier et al., 2018) is available here: https://doi.org/10.5067/NX1R09ORNOZN, and the Sentinel-1 data (Copernicus, 2020) is available here: https://asf.alaska.edu/data-sets/sar-data-sets/sentinel-1/.

*Video supplement.* The supplemental videos provide a NASA Worldview animation, and TSE buoy animation of the Beaufort Sea ice breakup detailed in Section 3.3. The animations can be viewed at https://youtu.be/WHCsOaL4Nks and https://youtu.be/l2tOJSnTfSY .

*Author contributions.* NA and RS designed and developed the experiment. NA and JL developed the methods. NA and JH contributed calculations for the analysis. All authors contributed to the analysis of the results and the writing of the manuscript.

*Competing interests.* The authors declare no competing interests.

*Acknowledgements.* The authors would like to thank Harry Heorton, Marcello Vichi, and one anonymous reviewer for their suggestions that significantly improved the manuscript. The authors would like to acknowledge funding from the Swiss National Science Foundation Postdoc Mobility Fellowship Project P400P2 199190 for Nikolas Aksamit, and a Natural Sciences and Engineering Research Council of Canada (NSERC) Discovery Grant for Randall Scharien. Jennifer Hutchings was funded through the National Science Foundation grant 1722729. The authors would also like to thank Angela Bliss for preparing and providing the MOSAiC buoy data, who accepts acknowledgement instead of co-authorship. Data used in this manuscript was produced as part of the international Multidisciplinary drifting Observatory for the Study of the Arctic Climate (MOSAiC) with the tag MOSAiC20192020 and the Project ID: AWI_PS122_00. We thank all those who contributed to MOSAiC and made this endeavor possible (Nixdorf et al. 2021).

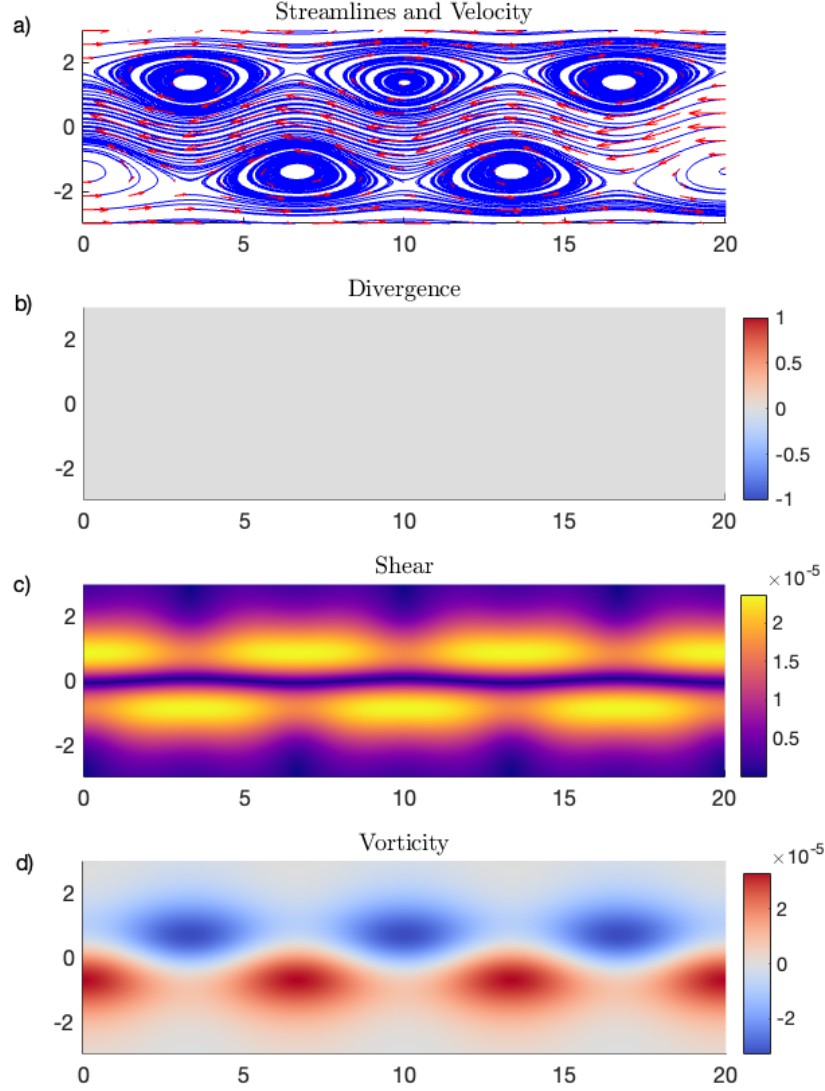

**Figure A1.** a) Streamlines and velocity vectors for the geophysical Bickley jet. b) The flow is designed to be diveregence-free, as can be verified by a uniform divergence value of zero. c) Instantaneous values of shear indicate strong shear zones adjacent to the top and bottom eddies and a narrow shear-free core. d) Vorticity fields correctly identify the centers of the eddies above and below the jet. Note that the contours of shear do not agree with the shape of eddies in a) and d).

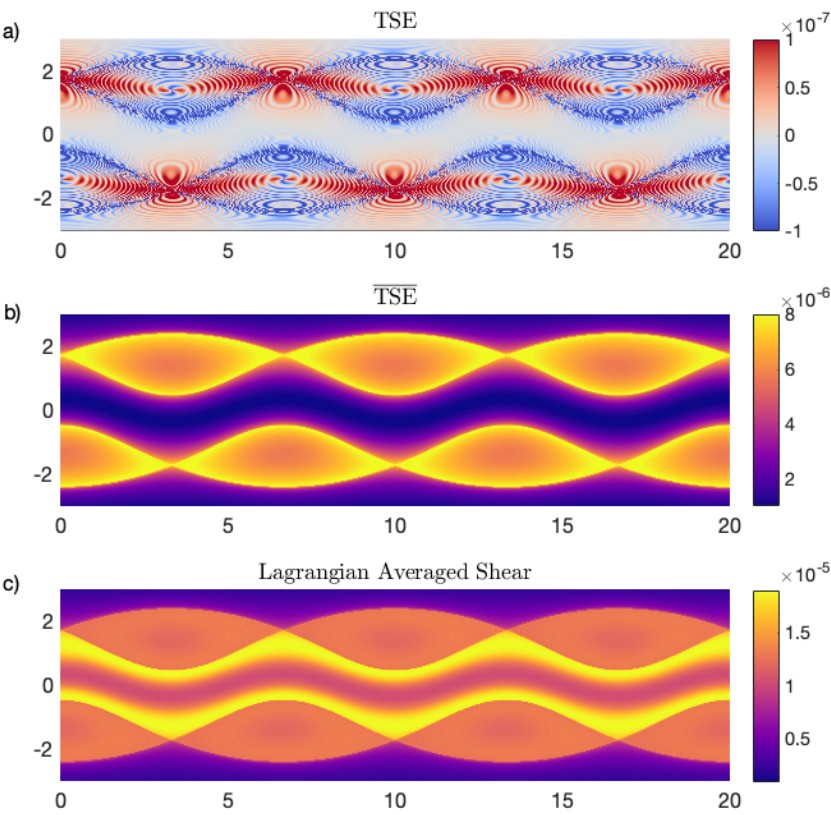

**Figure A2.** a) TSE values show a complex stretch-relax pattern in the Bickley jet, but still accurately identify the edges of each eddy as well as the central jet. b) $\overline{\text{TSE}}$ contours clearly reveal the central jet and the adjacent eddies. There is limited trajectory-tangent stretching in the central jet, with a slight increase near the boundaries. c) Lagrangian averaged shear also reveals the multiple eddies and central jet, in contrast to Eulerian shear measurements. Lagrangian averaged divergence is omitted as it is everywhere zero.

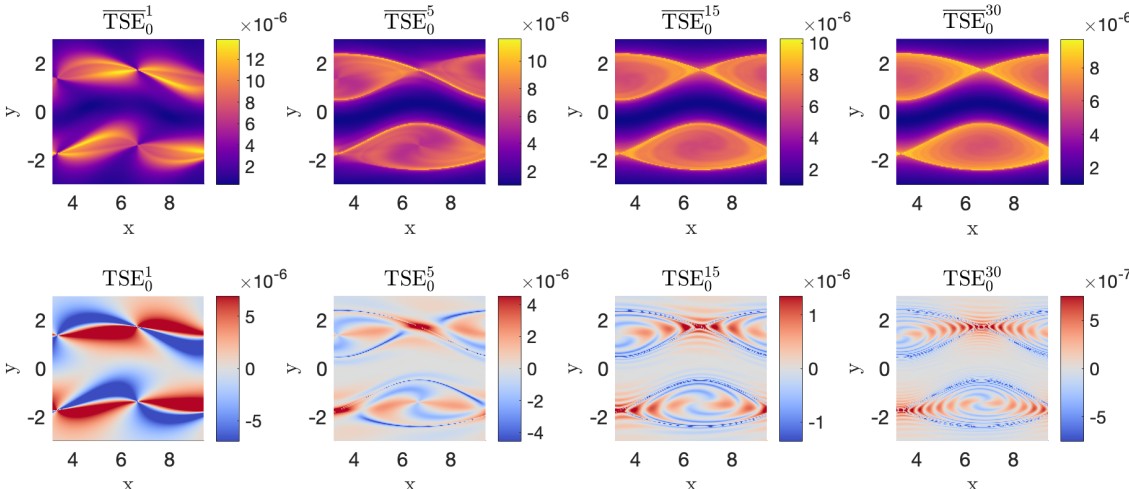

**Figure A3.** Comparison of TSE and $\overline{\text{TSE}}$ contours for increase integration times from 1 to 30 days (L to R) for a subsection of the Bickley jet. The eddies, saddles between eddies, and central jet are evident for all integration times, but the clarity of boundaries increases as integration time increases. Note this improvement is not always possible with time-dependent flows as features may change.

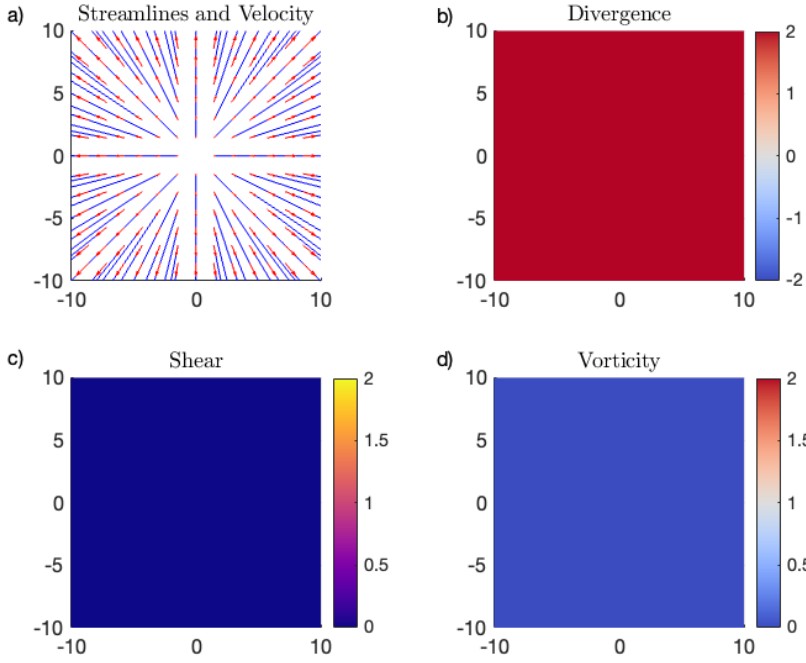

**Figure A4.** a) Streamlines and velocity vectors for the divergent flow (A13). b) The flow is designed to be uniformly divergent divergence value of 2. c) Instantaneous values of shear no shear. d) Vorticity fields are also uniformly zero.

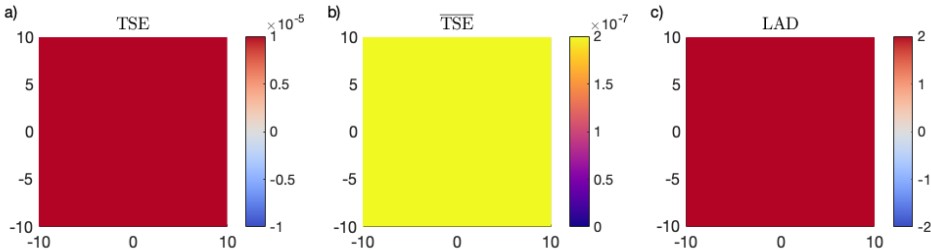

**Figure A5.** a-b) TSE and $\overline{\text{TSE}}$ show uniform stretching due to the uniform linear acceleration away from the origin. This matches the structure revealed by the uniform divergence field. c) Lagangrian averaged divergence is also constant, thus presenting the same pattern as TSE and $\overline{\text{TSE}}$. $LS$ is omitted as it is constantly zero in this shear-free flow.

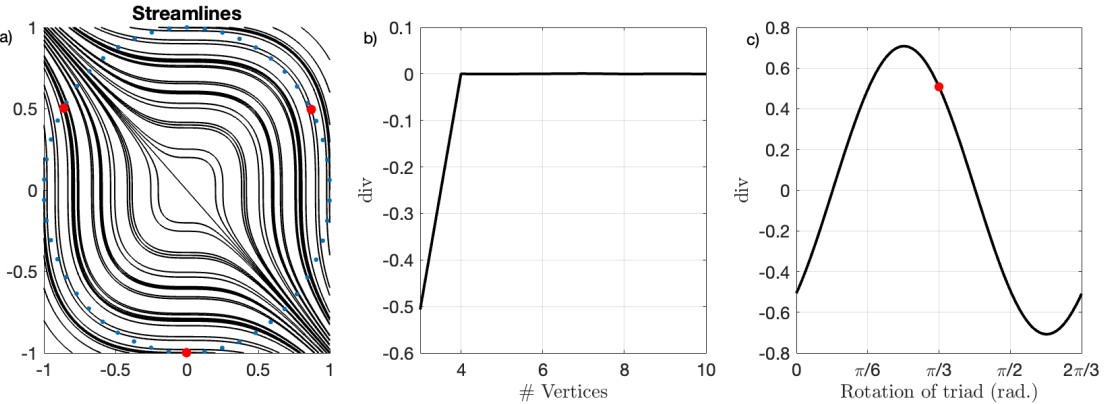

**Figure A6.** a) Streamlines for the flow in (A14) with one example triad position (red) and unit circle of rotation (blue). b) Average divergence as approximated by unit polygons as a function of the number of vertices used in the approximation. c) Average divergence as approximated by unit equilateral triad for all possible rotations with divergence correspond to panel a) triad in red.

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

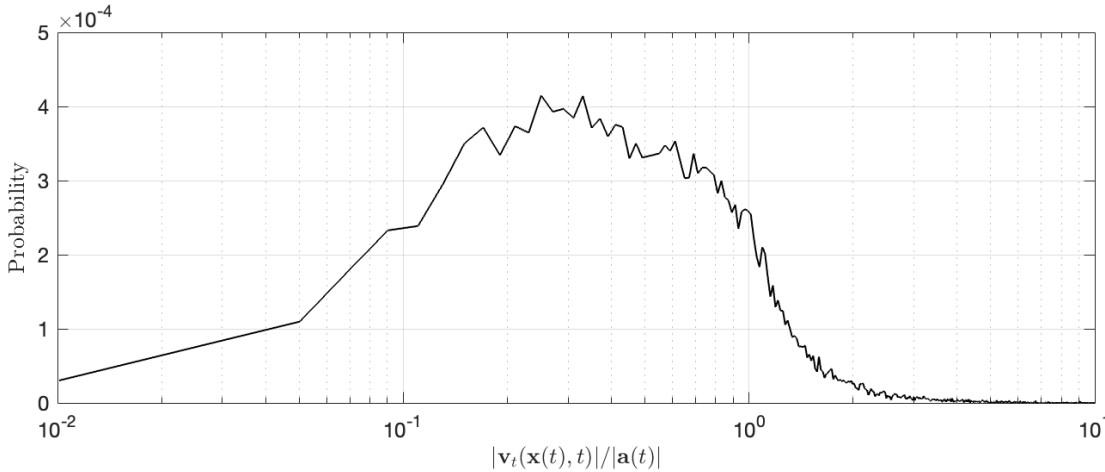

**Figure A7.** Example assessment of the slowly-varying assumption of Arctic sea ice velocities from Polar Pathfinder Daily 25 km EASE-Grid Sea Ice Motion Vectors, Version 4. $|\mathbf{v}_t(\boldsymbol{x}(t),t)|/|\mathbf{a}(t)|$ calculated from 50 days of gridded sea ice trajectories surrounding the Figure 9 analysis in 2017.

Bouillon, S. and Rampal, P.: On producing sea ice deformation data sets from SAR-derived sea ice motion, Cryosphere, 9, 663–673, https://doi.org/10.5194/tc-9-663-2015, 2015.

Cohen, L., Hudson, S. R., Walden, V. P., Graham, R. M., and Granskog, M. A.: Meteorological conditions in a thinner Arctic sea ice regime from winter to summer during the Norwegian Young sea ice expedition (N-ICE2015), Journal of Geophysical Research, 122, 7235–7259, https://doi.org/10.1002/2016JD026034, 2017.

Copernicus: Sentinel data 2020. Retrieved from ASF DAAC 17 March, 2022, processed by ESA, 2020.

Dai, A., Luo, D., Song, M., and Liu, J.: Arctic amplification is caused by sea-ice loss under increasing CO 2, Nature Communications, 10,

1–13, https://doi.org/10.1038/s41467-018-07954-9, 2019.

Dierking, W., Stern, H. L., and Hutchings, J. K.: Estimating statistical errors in retrievals of ice velocity and deformation parameters from satellite images and buoy arrays, Cryosphere, 14, 2999–3016, https://doi.org/10.5194/tc-14-2999-2020, 2020.

Encinas-Bartos, A. P., Aksamit, N. O., and Haller, G.: Quasi-objective eddy visualization from sparse drifter data, Chaos, 32, 113 143, https://doi.org/10.1063/5.0099859, 2022.

Graham, R. M., Cohen, L., Ritzhaupt, N., Segger, B., Graversen, R. G., Rinke, A., Walden, V. P., Granskog, M. A., and Hudson, S. R.: Evaluation of six atmospheric reanalyses over Arctic sea ice from winter to early summer, Journal of Climate, 32, 4121–4143, https://doi.org/10.1175/JCLI-D-18-0643.1, 2019a.

Graham, R. M., Itkin, P., Meyer, A., Sundfjord, A., Spreen, G., Smedsrud, L. H., Liston, G. E., Cheng, B., Cohen, L., Divine, D., Fer, I., Fransson, A., Gerland, S., Haapala, J., Hudson, S. R., Johansson, A. M., King, J., Merkouriadi, I., Peterson, A. K., Provost, C., Randelhoff,

A., Rinke, A., Rösel, A., Sennéchael, N., Walden, V. P., Duarte, P., Assmy, P., Steen, H., and Granskog, M. A.: Winter storms accelerate the demise of sea ice in the Atlantic sector of the Arctic Ocean, Scientific Reports, 9, 1–16, https://doi.org/10.1038/s41598-019-45574-5, 2019b.

Granskog, M. A., Rösel, A., Dodd, P. A., Divine, D., Gerland, S., Martma, T., and Melanie J. Leng: Snow contribution to first-year and second-year Arctic sea ice mass balance north of Svalbard, Journal of Geophysical Research: Oceans, 122, 1–22, https://doi.org/10.1002/2016JC012398.Received, 2017.

Grenfell, T. C., Barber, D. G., Fung, A. K., Gow, A. J., Jezek, K. C., Knapp, E. J., Nghiem, S. V., Onstott, R. G., Perovich, D. K., Roesler, C. S., Swift, C. T., and Tanis, F.: Evolution of electromagnetic signatures of sea ice from initial formation to the establishment of thick first-year ice, IEEE Transactions on Geoscience and Remote Sensing, 36, 1642–1654, https://doi.org/10.1109/36.718636, 1998.

Gurtin, M. E.: An Introduction to Continuum Mechanics, Academic Press, San Diego, USA, 1st edn., 1981.

Haller, G.: Lagrangian Coherent Structures, Annual review of fluid mechanics, pp. 137–162, https://doi.org/10.1002/9783527639748.ch3, 2015.

Haller, G. and Yuan, G.: Lagrangian coherent structures and mixing in two-dimensional turbulence, Phys. D Nonlinear Phenom., 147, 352–370, https://doi.org/10.1016/S0167-2789(00)00142-1, 2000.

Haller, G., Aksamit, N. O., and Bartos, A. P. E.: Quasi-Objective Coherent Structure Diagnostics from Single Trajectories, Chaos, 31, 043 131–1–17, https://doi.org/10.1063/5.0044151, 2021.

Haller, G., Aksamit, N., and Encinas-Bartos, A. P.: Erratum:"Quasi-objective coherent structure diagnostics from single trajectories" [Chaos 31, 043131 (2021)], Chaos, 32, 059 901, https://doi.org/10.1063/5.0090124, 2022.

Hutchings, J. K., Roberts, A., Geiger, C. A., and Richter-Menge, J.: Spatial and temporal characterization of sea-ice deformation, Annals of Glaciology, 52, 360–368, https://doi.org/10.3189/172756411795931769, 2011.

Hutchings, J. K., Heil, P., Steer, A., and Hibler, W. D.: Subsynoptic scale spatial variability of sea ice deformation in the western Weddell Sea during early summer, Journal of Geophysical Research: Oceans, 117, https://doi.org/10.1029/2011JC006961, 2012.

International Arctic Buoy Programme: International Arctic Buoy Programme, digital media, available at: http://iabp.apl.washington.edu/index.html (last access: March 2022), 2022.

Itkin, P., Spreen, G., Cheng, B., Doble, M., Gerland, S., and Granskog, M. A. . . . Helgeland, C.: N-ICE2015 buoy data. Norwegian Polar Institute., https://doi.org/10.21334/npolar.2015.6ed9a8ca, 2015.

Itkin, P., Haapala, J., Spreen, G., Cheng, B., Doble, M., Girard-Arduin, F., Hughes, N., Kaleschke, L., Nicolaus, M., and Wilkinson, J.: Thin ice and storms: Sea ice deformation from buoy arrays deployed during N-ICE2015, Journal of Geophysical Research: Oceans, 122, 1–22, https://doi.org/10.1002/2016JC012403.Received, 2017.

Jenkins, M. and Dai, A.: The Impact of Sea-Ice Loss on Arctic Climate Feedbacks and Their Role for Arctic Amplification, Geophysical Research Letters, 48, 1–9, https://doi.org/10.1029/2021GL094599, 2021.

Knust, R.: Polar Research and Supply Vessel POLARSTERN operated by the Alfred-Wegener-Institute, Journal of large-scale research facilities JLSRF, 3, 1–8, https://doi.org/10.17815/jlsrf-3-163, 2017.

Krumpen, T., Birrien, F., Kauker, F., Rackow, T., Von Albedyll, L., Angelopoulos, M., Jakob Belter, H., Bessonov, V., Damm, E., Dethloff, K., Haapala, J., Haas, C., Harris, C., Hendricks, S., Hoelemann, J., Hoppmann, M., Kaleschke, L., Karcher, M., Kolabutin, N., Lei, R., Lenz, J., Morgenstern, A., Nicolaus, M., Nixdorf, U., Petrovsky, T., Rabe, B., Rabenstein, L., Rex, M., Ricker, R., Rohde, J., Shimanchuk, E., Singha, S., Smolyanitsky, V., Sokolov, V., Stanton, T., Timofeeva, A., Tsamados, M., and Watkins, D.: The MOSAiC ice floe: Sediment-laden survivor from the Siberian shelf, Cryosphere, 14, 2173–2187, https://doi.org/10.5194/tc-14-2173-2020, 2020.

Krumpen, T., Von Albedyll, L., Goessling, H. F., Hendricks, S., Juhls, B., Spreen, G., Willmes, S., Belter, H. J., Dethloff, K., Haas, C., Kaleschke, L., Katlein, C., Tian-Kunze, X., Ricker, R., Rostosky, P., Rückert, J., Singha, S., and Sokolova, J.: MOSAiC drift expedi-

tion from October 2019 to July 2020: Sea ice conditions from space and comparison with previous years, Cryosphere, 15, 3897–3920,
https://doi.org/10.5194/tc-15-3897-2021, 2021.

Kwok, R. and Rothrock, D. A.: Decline in Arctic sea ice thickness from submarine and ICESat records: 1958-2008, Geophysical Research
Letters, 36, 1–5, https://doi.org/10.1029/2009GL039035, 2009.

Kwok, R., Curlander, J. C., Pang, S. S., and Mcconnell, R.: An Ice-Motion Tracking System at the Alaska SAR Facility, IEEE Journal of
Oceanic Engineering, 15, 44–54, https://doi.org/10.1109/48.46835, 1990.

Landy, J. C., Dawson, G. J., Tsamados, M., Bushuk, M., Stroeve, J. C., Howell, S. E., Krumpen, T., Babb, D. G., Komarov, A. S., Heo-
rton, H. D., Belter, H. J., and Aksenov, Y.: A year-round satellite sea-ice thickness record from CryoSat-2, Nature, 609, 517–522,
https://doi.org/10.1038/s41586-022-05058-5, 2022.

Lei, R., Hoppmann, M., Cheng, B., Zuo, G., Gui, D., Cai, Q., Jakob Belter, H., and Yang, W.: Seasonal changes in sea ice kinematics and
deformation in the Pacific sector of the Arctic Ocean in 2018/19, Cryosphere, 15, 1321–1341, https://doi.org/10.5194/tc-15-1321-2021,
2021.

Lei, R., Cheng, B., Hoppmann, M., Zhang, F., Zuo, G., Hutchings, J. K., Lin, L., Lan, M., Wang, H., Regnery, J., Krumpen, T., Haapala, J.,
Rabe, B., Perovich, D. K., and Nicolaus, M.: Seasonality and timing of sea ice mass balance and heat fluxes in the Arctic transpolar drift
during 2019- 2020, Elementa, 10, 1–22, https://doi.org/10.1525/elementa.2021.000089, 2022.

Leppäranta, M.: The Drift of Sea Ice, Springer-Verlag, second edn., 2011.

Lindsay, R. W. and Stern, H. L.: The RADARSAT Geophysical Processor System: Quality of sea ice trajectory and de-
formation estimates, Journal of Atmospheric and Oceanic Technology, 20, 1333–1347, https://doi.org/10.1175/1520-
0426(2003)020<1333:TRGPSQ>2.0.CO;2, 2003.

Lukovich, J. V., Bélanger, C., Barber, D. G., and Gratton, Y.: On coherent ice drift features in the southern Beaufort sea, Deep-Sea Research
Part I: Oceanographic Research Papers, 92, 56–74, https://doi.org/10.1016/j.dsr.2014.05.013, 2014.

Lukovich, J. V., Hutchings, J. K., and Barber, D. G.: On sea-ice dynamical regimes in the Arctic ocean, Annals of Glaciology, 56, 323–331,
https://doi.org/10.3189/2015AoG69A606, 2015.

Lukovich, J. V., Geiger, C. A., and Barber, D. G.: Method to characterize directional changes in Arctic sea ice drift and associated deformation
due to synoptic atmospheric variations using Lagrangian dispersion statistics, Cryosphere, 11, 1707–1731, https://doi.org/10.5194/tc-11-
720    1707-2017, 2017.

Lukovich, J. V., Stroeve, J. C., Crawford, A., Hamilton, L., Tsamados, M., Heorton, H., and Massonnet, F.: Summer extreme cyclone impacts
on arctic sea ice, Journal of Climate, 34, 4817–4834, https://doi.org/10.1175/JCLI-D-19-0925.1, 2021.

Meier, W. N., Comiso, J. C., and Markus, T.: AMSR-E/AMSR2 Unified L3 Daily 6.25 km Polar Gridded 89 GHz Brightness Temperatures,
Version 1., https://doi.org/https://doi.org/10.5067/NX1R09ORNOZN, 2018.

Moore, G. W., Schweiger, A., Zhang, J., and Steele, M.: Collapse of the 2017 Winter Beaufort High: A Response to Thinning Sea Ice?,
Geophysical Research Letters, 45, 2860–2869, https://doi.org/10.1002/2017GL076446, 2018.

Nicolaus, M., Perovich, D. K., Spreen, G., Granskog, M. A., von Albedyll, L., Angelopoulos, M., Anhaus, P., Arndt, S., Jakob Belter, H.,
Bessonov, V., Birnbaum, G., Brauchle, J., Calmer, R., Cardellach, E., Cheng, B., Clemens-Sewall, D., Dadic, R., Damm, E., de Boer, G.,
Demir, O., Dethloff, K., Divine, D. V., Fong, A. A., Fons, S., Frey, M. M., Fuchs, N., Gabarró, C., Gerland, S., Goessling, H. F., Gradinger,
R., Haapala, J., Haas, C., Hamilton, J., Hannula, H. R., Hendricks, S., Herber, A., Heuzé, C., Hoppmann, M., Høyland, K. V., Huntemann,
M., Hutchings, J. K., Hwang, B., Itkin, P., Jacobi, H. W., Jaggi, M., Jutila, A., Kaleschke, L., Katlein, C., Kolabutin, N., Krampe, D.,
Kristensen, S. S., Krumpen, T., Kurtz, N., Lampert, A., Lange, B. A., Lei, R., Light, B., Linhardt, F., Liston, G. E., Loose, B., Macfarlane,

A. R., Mahmud, M., Matero, I. O., Maus, S., Morgenstern, A., Naderpour, R., Nandan, V., Niubom, A., Oggier, M., Oppelt, N., Pätzold, F., Perron, C., Petrovsky, T., Pirazzini, R., Polashenski, C., Rabe, B., Raphael, I. A., Regnery, J., Rex, M., Ricker, R., Riemann-Campe, K., Rinke, A., Rohde, J., Salganik, E., Scharien, R. K., Schiller, M., Schneebeli, M., Semmling, M., Shimanchuk, E., Shupe, M. D., Smith, M. M., Smolyanitsky, V., Sokolov, V., Stanton, T., Stroeve, J., Thielke, L., Timofeeva, A., Tonboe, R. T., Tavri, A., Tsamados, M., Wagner, D. N., Watkins, D., Webster, M., and Wendisch, M.: Overview of the MOSAiC expedition: Snow and sea ice, Elem Sci Anth, 10, https://doi.org/10.1525/elementa.2021.000046, 2022.

Nolan, P. J., Serra, M., and Ross, S. D.: Finite-time Lyapunov exponents in the instantaneous limit and material transport, Nonlinear Dynamics, 100, 3825–3852, https://doi.org/10.1007/s11071-020-05713-4, 2020.

Oikkonen, A., Haapala, J., Lensu, M., Karvonen, J., and Itkin, P.: Small-scale sea ice deformation during N-ICE2015: From compact pack ice to marginal ice zone, Journal of Geophysical Research: Oceans, 122, 5105–5120, https://doi.org/10.1002/2016JC012387.Received, 2017.

Ott, W. and Yorke, J. A.: When Lyapunov exponents fail to exist, Physical Review E - Statistical, Nonlinear, and Soft Matter Physics, 78, 1–6, https://doi.org/10.1103/PhysRevE.78.056203, 2008.

Rabe, B., Heuze, C., Regnery, J., Aksenov, Y., Allerholt, J., Athanase, M., and Bai, Youcheng ... Zhu, J.: Overview of the MOSAiC expedition : Physical oceanography, Elem Sci Anth, 10, 1–31, https://doi.org/10.1525/elementa.2021.00062, 2022.

Rampal, P., Weiss, J., Marsan, D., and Bourgoin, M.: Arctic sea ice velocity field: General circulation and turbulent-like fluctuations, Journal of Geophysical Research: Oceans, 114, 1–17, https://doi.org/10.1029/2008JC005227, 2009.

Rampal, P., Dansereau, V., Olason, E., Bouillon, S., Williams, T., Korosov, A., and Samaké, A.: On the multi-fractal scaling properties of sea ice deformation, Cryosphere, 13, 2457–2474, https://doi.org/10.5194/tc-13-2457-2019, 2019.

Rothrock, D. A., Yu, Y., and Maykut, G. A.: Thinning of the Arctic Sea-Ice cover, Geophysical Research Letters, 26, 3469–3472, https://doi.org/10.1029/1999GL010863, 1999.

Rypina, I. I., Brown, M. G., Beron-Vera, F. J., Koçak, H., Olascoaga, M. J., and Udovydchenkov, I. A.: On the Lagrangian Dynamics of Atmospheric Zonal Jets and the Permeability of the Stratospheric Polar Vortex, Journal of the Atmospheric Sciences, 64, 3595–3610, https://doi.org/10.1175/JAS4036.1, 2007.

Screen, J. A. and Simmonds, I.: The central role of diminishing sea ice in recent Arctic temperature amplification, Nature, 464, 1334–1337, https://doi.org/10.1038/nature09051, 2010.

Serra, M. and Haller, G.: Objective eulerian coherent structures, Chaos, 26, https://doi.org/10.1063/1.4951720, 2016.

Siew, P. Y. F., Li, C., Sobolowski, S. P., and King, M. P.: Intermittency of Arctic–mid-latitude teleconnections: stratospheric pathway between autumn sea ice and the winter North Atlantic Oscillation, Weather and Climate Dynamics, 1, 261–275, https://doi.org/10.5194/wcd-1-261-2020, 2020.

Szanyi, S., Lukovich, J. V., and Barber, D. G.: Lagrangian analysis of sea-ice dynamics in the Arctic Ocean, Polar Research, 35, https://doi.org/10.3402/polar.v35.30778, 2016a.

Szanyi, S., Lukovich, J. V., Barber, D. G., and Haller, G.: Persistent artifacts in the NSIDC ice motion data set and their implications for analysis, Geophysical Research Letters, 43, 10,800–10,807, https://doi.org/10.1002/2016GL069799, 2016b.

Thackeray, C. W. and Hall, A.: An emergent constraint on future Arctic sea-ice albedo feedback, Nature Climate Change, 9, 972–978, https://doi.org/10.1038/s41558-019-0619-1, 2019.

Truesdell, C. and Noll, W.: The Non-Linear Field Theories of Mechanics, Springer, 3rd edn., https://doi.org/10.1007/978-3-662-10388-3_1, 2004.

Tschudi, M., Meier, W. N., Stewart, J. S., Fowler, C., and Maslanik, J.: Polar Pathfinder Daily 25 km EASE-Grid Sea Ice Motion Vectors, Version 4. Accessed: March 3, 2022., https://doi.org/https://doi.org/10.5067/INAWUWO7QH7B, 2018.