# Peer review of "A quasi-objective single buoy approach for understanding Lagrangian coherent structures and sea ice dynamics"

_EGUsphere, 2022_

## Referee Comment (RC2)

This paper introduces a new method for obtaining information on sea ice dynamics from floating ice tethered buoy location data. The theory is introduced and implemented on n3 separate buoy data sets. The study presents an interesting and worthwhile inclusion to current methods of analysing sea ice drift. Comparisons are successfully made in the paper that show notable improvements overt existing polygon methods of analysing sea ice drift and dynamics. However, I find the paper somewhat lacking in results and technical information and this greatly hampers the interpretation of the results presented. I am confident that after addressing the following issues it will be fit for publication. See major points below followed by specific points.

A more detailed description of the usability of this method is needed. This could well be a useful method for the community to explore sea ice dynamics, but certain features of the method and results are still unclear to me.

First is the dimensionality of the results. The figures give results in units of d-1, so it seems that TSE metrics are equivalent to standard strain measurements. How do the units given in the results scale in comparison to Itkin (2017) and other measurements? Are the TSE scales and strain measurements of comparable magnitude? Is TSE most comparable to divergence, how does it respond to shearing across the trajectory? A few toy examples of this method, such as included in the appendix will aid this paper. For example, it is currently unclear to me what TSE we expect for a large coherent ice cover, under no deformation, but under acceleration. What TSE do we expect for a divergent flow? How does it respond under rotation or with a flow field experiencing shear or curl? These examples will aid the interpretation of particularly figure 2, where the TSE and triangle methods show different behaviour. What is happening to the ice during these periods?

The second is that the method relies on the magnitude of the tangential vector to the buoy trajectory. Does this mean that directional information is included within the TSE results, or is it purely a scalar? Does this link into the analysis between the TSE and polygon methods on L 199? The context of figure 1 is difficult to understand. This may be due to the difficulty in interpreting the method and theory presented, but consider adding to this figure the deformation results of Itkin (2017), if comparable. The figure captions all need expanding upon.

Thirdly a greater description of \bar{TSE} is required. This is given as the "hyperbolicity strength" but this definition does not explain to me why this value is positive definite. The equational form and results suggest that this metric gives a longer time scale measurement of stretching, but only for divergence with no expression of compression. In figure 2 it is compared to the strain rate magnitudes, so is it a measure of the total magnitude of change in stretching?

Fourth, the buoy deformation results in figure 2 are unexplained and uncited. Are these the first publication of the triangle based deformation measurements from MOSAIC? If so they need much more documentation than currently included here and possibly a figure or two to allow these results to be adequately interpreted. If not then a further description of the previous publication is required. How reliable are the results? What are the successes of these results?
Related to this issue: a separate data section is required. This needs to include all descriptions of the data used and the previous results repeated in this study. Currently this information is within the introduction, method and results and is difficult to follow.

L 14 a more up to date reference for this is desirable.

L 15 -16. Does this feedback come directly from Serreze and Francis? A little more expansion on how they discovered and documented is needed. The current description is too brief to show the importance of ice dynamics.

L 28-29 this sentence doesn't fit the flow of the paragraph. Consider moving it before the description of SAR data.

L 35 This paragraph will benefit from an expanded definition of a Lagrangian coherent structure, in particular why this perspective results in the difficulties in using 'gridded sea ice displacement fields' mentioned later.

L 46 What characteristics of a LCS make it a hyperbolic LCS? And what makes sea ice applicable to a hyperbolic LCS? Additional arguments and descriptions of Haller et al. 2021 could be incorporated.

L 50 Does the period have the 'much larger influence' or is it the identification that has it? 'Much larger influence' than what?

L 63 'the the' -> 'that the'

L 63, 64 I find that this sentence does not give enough background on Haller et al. 2021 to allow for any understanding of the following equations. Please include a sentence each, with terms, on 'material stretching', 'hyperbolicity strength' and 'initial material tangent vector'. At the moment the reader is required to also read a large part of Haller et al. 2021 in order to understand these equations. It is also unclear what is represented in the two equations.

L 69 My interpretation of the appendix does not show that this is "verified". If I indeed it is only likely that sea ice drift is slowly varying, then this needs to be stated as such.

L 78 including a definition of a "steady flow" will benefit this section.

L 87 Again a definition of hyperbolicity in this context will aid the understanding here.

L 90 Can you add in this paragraph a description on the units of the two equations and TSE? A further description of how this relates to usual deformation units and how to interpret the two values would be of help (you may want to put this elsewhere).

L 91 A citation is needed here.

L 95 - 97 What is meant by this sentence? Will this technique be used later, or is it a note on the context of TSE methods and the use of existing stress vs train rheology methodologies?

L 112 It is not immediately obvious why equations 6-9 are included as they are not referenced. Consider removing them. Do they apply directly to the example A3? If so put them there too. The following paragraph gives a detail dsicussion on the limitations of polygon based approaches, these extra questions don't bring anything useful here.

L 123 An example of such a long time series is needed here.

L 149 An extra summary sentence here showing plain words rational for this technique would be beneficial.

L 169 how is the slowly varying nature of sea ice drift relate to these storm periods? Is it more or less likely that the slowly vary criterion holds?

L 175 I'm not sure why the beginning part of this sentence is needed, as the second part of it does not logically follow. It's fine that the diagnostic has a time window, and this description is a sensible choice.

L 179 here may be good place to refer to equations 6-9, if they are needed at all.

L 186 a quick summary of the Itkin cleaning method would be a beneficial addition here.

Figure 1. This caption requires extensive expansion. All lines need to be defined. It is currently impossible to interpret this figure without extensive reading in the text. The figure needs to be interpretable from the caption alone assuming a knowledge on the papers aims and method. Addional lines at d-1 = 0 will allow

L 217 - 221 Has this data been analysed by this method previously? If so citations and a summary of results is required. If not then this paper needs expansion as a presentation of these new results too. A least a discussion of previous use of these results or method is required.

L225 please refer div, D back to the equations previously and change the labels on the plot to directly match the text. Using 2a, 2b will help too. Please also add from an improved method why TSE is compared to div, and \bar{TSE} to total D.

L 240 which source do these numbers come from? The polygon of triangle based methods? What numbers come from the other method? Can the two methods be dimensionally compared in this way? Is there any method that allows for the integral of all deformation and TSE over the period discussed?

L 241 Is this value significant? Which line plot does it come from? Do all significant deformations have a higher value?

L 242 Is shear plotted anywhere? How do we interpret shear against the TSE metrics?

Figure 2. Caption needs expanding. (a-d) are referenced in the text but do not appear. What is the black line in b and d? A scale is required for the colourbar.

L 258 Please comment on how the spacing of the buoys and the time handling of this data (linear sampling) affects the dimensionality of the calculated TSE in comparison to high time resolution data from the other sources.

L 259 Shown in black where?

L 267 In the line plot I see that at the beginning of the period the TSE is distributed about zero, and then towards these events the spread of values reduces to oscillating peaks. Is this what you mean?

L 270 Red is positive TSE? So equivalent to net divergence?

L271 high positive or negative values?

L 275 This paragraph will be aided by a previous discussion of what TSE we expect for certain dynamics events. For the accelerating ice described here, what TSE is expected? For constant but rotating flow as described at the end of this paragraph what idealised TSE is expected?

Figure 3 You have chose not to include \bar{TSE} int his plot. Can you explain why? Please use a divergent colour scale for the divergence, with white at zero and different colours for positive/negative TSE.

L 281 This paper has provided not explantation I could interpret so far on how TSE provides insight in to "distant fracture events". Please expand, as this is a useful contribution if true.

L 285 I have seen the deformation events captured in the new technique, but nothing on prediction. Perhaps the wrong word to use here.

L 291 "predicted" again this is the wrong word I think. Captured or similar is more accurate.

L 296 "Buildup of stress" is this your hypothesised stress state prior to the break up, or does the TSE measure stress? Are there other measurements of stress during this period that can back up this claim?

L298 Again is this a hypothesis of what internal stresses are expected within the pack? If these assessments of ice stress are speculation, please be very clear about this, or remove them.

L 317 This is the first mention of the technical methodology used in this study. Please include this information earlier in the study also. A data section detailing the exact values taken from the buoy data is needed.

Appendices

L 362-365 Can you comment on how figure A1 suggests that the slowly varying criterion is met, but not conclusively? I see from figure A1 that $|vt|/|a(t)| < 1$ in most cases (positive tail greater than $10^0$), but not strictly or $\ll 1$. Is this correct? Does that mean that the data presented suggests that sea ice is, in general, slowly varying, but not strictly so? Can you comment on the cases where $|vt|/|a(t)| > 1$, when do such cases occur?

L 394 A summary sentence for this example, repeating, and expanding upon the introduction to the appendix would be helpful here.

---

## Author Comment (AC1)

Thank you for taking the time to review our manuscript. We have responded to your comments below.

General comments:

**Slowly-varying condition of sea ice drifting is the main basis for the application of TSE method. Then, is the TSE method applicable to all sea ice concentration scenarios and reasonable for all cases with different ice-wind ratios? That is, how the internal stress of sea ice affects the method. It is suggested that the author strengthen the discussion in this respect and increase the influence of sea ice concentration on the TSE calculation results.**

*Thank you for bringing this to our attention. We have further expanded our methods section to explain what exactly TSEs are calculating. We also have added an appendix section that shows the derivation of TSEs and relates them to Shear and Divergence in a analytic model example. TSE calculations are affected by sea ice concentration in much the same way as array-based estimates of rate-of-strain metrics. Once we are no longer dealing with a continuum of ice, such as when ice is transitioning to free-drift, we are no longer representing ice deformation, but calculating motion in a ice-ocean mixture. We have added to our MOSAiC example what happens when sea ice concentration drops and our cluster of buoys is no longer correlated.*

**As the author said, the speed of sea ice motion is very dependent on the sampling frequency (See also Lei et al., 2021). Then, the three groups of buoy data used in the paper should have different sampling frequencies. What is the impact?**

*At lower sampling frequencies, we might lose high frequency oscillations in the ice. For TSE calculations, this effect would be negligible. For \bar{TSE}, this oscillation effect would be cumulative and could indicate much higher degree of absolute hyperbolicity (cumulative stretching and compression). For example, a low sampling frequency might miss tidal oscillations, whereas higher frequency would be able to identify this effect on the ice. We now explain in the data section how the choice of sampling frequency is derived from the data source or comparisons with previous experiments.*

*In essence we are trying to approximate a continuous integral, and the sampling period is defining the discrete sum we are using to approximate that integral. I appreciate the concern of the reviewer with respect to sampling rate, but this calculus problem is well studied, and unfortunately unavoidable with discrete data. Obviously, higher sampling rate is better if there are not additional errors included. Changes from subsampling depend on the interpolation approach.*

**In addition, when judging the slowly-varying nature of the sea ice flow, daily satellite remote sensing products are used. Although the author already mentioned its influence, I think it is necessary to give the degree of influence quantitatively.**

*Unfortunately, we cannot ascertain the impact of a daily output of the ice product until a comparable higher resolution ice product is available. To quantify the slowly-varying nature of the flow (as in the appendix), we need to calculate spatial and temporal derivatives, which is only available with a gridded motion product. Our understanding of the slowly-varying nature of sea ice would benefit from a concerted modeling study where we can vary the sampling frequency, which is now suggested in the text.*

Special comments:

- **Unit 20: "Obtaining local or regional information on the state of sea-ice can thus give an indication of future sea ice melt rates and potential weather impacts"**

**-- the Connector for sea-ice is not necessary. "future" is better change to summer because it is mainly about the seasonal scale.**

*Thank you. This has been changed to summer.*

- **Unit 80: "With these consideration in mind, we focus on mid-winter and early spring ice dynamics prior to minimize extensive fracturing of the ice cover"**

**-- Does this mean that this method is not applicable in the sea ice marginal ice zone or the area with low ice concentration in summer.**

*Trajectory stretching exponents measure the degree of stretching of a specific initial vector in a material as it evolves over time. If the material is discontinuous (such as in the marginal ice zone), you would no longer be measuring deformation of just the material, but stretching of a motion vector of these mixed continuua. This is the same as quantifying shear or divergence in the marginal ice zone. Values no longer represent shearing of solely the ice, but quantify a deformation motion in the ice-ocean continuum. Please also refer to our response to general comment #1.*

**Unit 155: Here is a paper (Lei et al., 2022) talks about the timing of sea ice mass balance at the MOSAiC DN. Although this is a process analysis of thermodynamics, I think seasonal thermodynamic processes are helpful for supporting the analysis of their kinematic and dynamic processes.**

*Thank you for your suggestion. We will integrate this reference in the next version of the manuscript.*

**Unit 180: ", LKF formation", Abbreviations are not defined.**

*Corrected, thank you.*

**Unit 280 "We find that TSEs successfully identify signnificant local material deformation tangent to individual sea ice buoy trajectories"**

**-Sea ice deformation has obvious localization characteristics (Lei et al., 2021). The deformation given based on TES method should only identify the deformation and**

fragmentation along the sea ice trajectory. Therefore, for a designated area (e.g., MOSAiC DN region), to obtain the localization characteristics of sea ice deformation, it is still necessary to build a high-density buoy array, even if the TES measurement method is used.

*Thank you. We have added a comment in the conclusions that it is still necessary to use a high-density buoy array to characterize sea ice deformation with rate-of-strain invariants, and to localize trajectory stretching at high resolution:*

*"To obtain rate-of-strain invariants for sea ice deformation, it is still necessary to use a high-density buoy array. Such an array also reveals gradients of trajectory stretching and further enhances precise stretching localization with TSEs."*

**Unit 295 "TSEs accurately predicted the onset of major storms"**

**Can you give the physical mechanism to explain this prediction. It is generally believed that sea ice deformation occurs during or after storms.**

*Thank you for pointing this out. We have clarified the sentence in line with the general consensus as follows:*

*"Specifically, large TSE values coincided with major storms in the N-ICE2015 experiment"*

**Data availability, The MOSAiC drifter data: The MOSAiC GPS buoys were jointly provided by colleagues participating in MOSAiC, so appropriate acknowledgements**

**are necessary. Because there were many providers involved, or sea ice team members can be used instead.**

*Thank you. The acknowledgements recognize Angela Bliss for her work in preparing and providing the MOSAiC buoy data.*

References:

Lei, R, et al. 2022. Seasonality and timing of sea ice mass balance and heat fluxes in the Arctic transpolar drift during 2019–2020. Elem Sci Anth, 10: 1. DOI: https://doi.org/10.1525/elementa.2021.000089.

Lei, R., Hoppmann, M., Cheng, B., Zuo, G., Gui, D., Cai, Q., Belter, H. J., and Yang, W.: Seasonal changes in sea ice kinematics and deformation in the Pacific sector of the Arctic Ocean in 2018/19, The Cryosphere, 15, 1321–1341, https://doi.org/10.5194/tc-15-1321-2021, 2021.

---

## Author Comment (AC2)

Thank you for taking the time to review our manuscript. We have responded to your comments below.

**A more detailed description of the usability of this method is needed. This could well be a useful method for the community to explore sea ice dynamics, but certain features of the method and results are still unclear to me.**

*Thank you for your suggestion. We have thoroughly expanded the methods section and added an additional appendix section to help the community understand these techniques.*

**First is the dimensionality of the results. The figures give results in units of d-1, so it seems that TSE metrics are equivalent to standard strain measurements. How do the units given in the results scale in comparison to Itkin (2017) and other measurements? Are the TSE scales and strain measurements of comparable magnitude? Is TSE most comparable to divergence, how does it respond to shearing across the trajectory? A few toy examples of this method, such as included in the appendix will aid this paper. For example, it is currently unclear to me what TSE we expect for a large coherent ice cover, under no deformation, but under acceleration. What TSE do we expect for a divergent flow? How does it respond under rotation or with a flow field experiencing shear or curl? These examples will aid the interpretation of particularly figure 2, where the TSE and triangle methods show different behaviour. What is happening to the ice during these periods?**

*Thank you for bringing these points to our attention. We have now included an additional section in the appendix where we clearly outline the mathematical connections of TSEs and divergence and shear by way of the rate-of-strain tensor.*

**The second is that the method relies on the magnitude of the tangential vector to the buoy trajectory. Does this mean that directional information is included within the TSE results, or is it purely a scalar?**

*TSE is purely a scalar as it is a measure of the rate stretching of trajectory-tangent vectors, the change in magnitudes.*

**Does this link into the analysis between the TSE and polygon methods on L 199?**

*I do not understand exactly what link you are referring to, but the differences in Green's theorem-based methods and TSE are thought to be the root of all the differences between our results and previous polygon-based findings. The effects on L199 may be from the choice of triads used, or Green's theorem approximation errors.*

**The context of figure 1 is difficult to understand. This may be due to the difficulty in interpreting the method and theory presented, but consider adding to this figure the deformation results of Itkin (2017), if comparable.**

*Thank you for your mention. We have expanded on the explanation and caption of Figure 1 to improve understanding. We find that adding the deformation results of Itkin would make the figure too busy, without adding much value. The connection we are drawing is with the*

*occurrence of storms that are important for sea ice dynamics, as the storms provide the external validation of TSEs. We do a more thorough comparison with array-based metrics in the MOSAiC example.*

**The figure captions all need expanding upon.**

*Thank you. The figure captions have all been expanded.*

**Thirdly a greater description of \bar{TSE} is required. This is given as the "hyperbolicity strength" but this definition does not explain to me why this value is positive definite.**

*Thank you for bringing this to our attention The value is positive because it is a sum of positive values. The term hyperbolicity strength derives from the original manuscript. A greater description has been included in the new methods section.*

**The equational form and results suggest that this metric gives a longer time scale measurement of stretching, but only for divergence with no expression of compression. In figure 2 it is compared to the strain rate magnitudes, so is it a measure of the total magnitude of change in stretching?**

*TSE is positive for stretching, and negative for compression along a trajectory. Thus, if the material stretches and then compresses to its original state, TSE = 0. \bar{TSE} does not allow for this cancelation, instead adding up all "hyperbolic"(stretching and compression) action. The term hyperbolic comes from the dynamical systems definition of hyperbolic manifolds that act as attracting and repelling structures, where nearby vectors undergo exceptional stretching or shrinking.*

**Fourth, the buoy deformation results in figure 2 are unexplained and uncited. Are these the first publication of the triangle based deformation measurements from MOSAIC? If so they need much more documentation than currently included here and possibly a figure or two to allow these results to be adequately interpreted. If not then a further description of the previous publication is required. How reliable are the results? What are the successes of these results?**

*The triad analysis we use in this paper for the MOSAiC data has not been previously published. A data paper for the buoys is currently under review (Bliss et al., 2022), and has now been referenced in the revised manuscript. The triad analysis was used in AGU 2020 and EGU 2021 presentations by Jenny Hutchings, and is chosen from buoy tracks that lasted for the full period from November 2019 deployment until June 2020. This is so we can create a time series for the full time without needing to account for changes in the array. We do have another paper we are working on that will improve upon this methodology by triangulating the array over shorter time periods, to provide a more detailed, potentially higher accuracy, and complete timeseries of total deformation and spatial variability in this deformation within the MOSAiC distributed network. However this paper is not ready for submission in the near future. The data we use here was chosen as it was created with a similar method to the past campaigns and that used for the IABP analysis. We do note that the MOSAiC buoy array is not well suited to automated triangulation methods. Delaunay triangulation creates skinny arrays that are less accurate for calculating deformation with. Hence similar to previous campaigns (SEDNA and ISPOL for*

*example) we hand-picked triangles within the array that ensured each triangle maintained as best a non-skewed shape as possible over the time period. This was achieved by checking triangle shapes by eye in November, March and June. We agree that it would be best to document this fully here, and have included figures that show the triads and their evolution during the time series. Please do note, this is not the definitive MOSAiC data set for sea ice deformation, it was simply chosen as a best representation of deformation to compare against the new method presented in this paper. We do not wish this paper to provide the definitive MOSAiC sea ice deformation time series for the triad method, but do believe that using a time series that was created with the method documented in Hutchings et al. (2012) is reasonable to show the utility of the new TSE method.*

**Related to this issue: a separate data section is required. This needs to include all descriptions of the data used and the previous results repeated in this study. Currently this information is within the introduction, method and results and is difficult to follow.**

*A new data section has been included.*

**L 14 a more up to date reference for this is desirable.**

*We have now included a more recent reference.*

**L 15 -16. Does this feedback come directly from Serreze and Francis? A little more expansion on how they discovered and documented is needed. The current description is too brief to show the importance of ice dynamics.**

*Thank you. The original reference was a bit confusing. This sentence has been changed to the follow*

*"As the ice warms in spring, melt is accelerated around existing fractures due to a reduction in albedo and the presence of more open water. Arctic amplification, the disproportionate warming of the arctic in a changing global climate, has been partially attributed to the enhanced oceanic heating and ice-albedo feedback caused by diminishing sea ice (Screen and Simmonds, 2010; Dai et al., 2019; Thackeray and Hall, 2019; Jenkins and Dai, 2021). "*

**L 28-29 this sentence doesn't fit the flow of the paragraph. Consider moving it before the description of SAR data.**

*Thank you. This has been relocated.*

**L 35 This paragraph will benefit from an expanded definition of a Lagrangian coherent structure, in particular why this perspective results in the difficulties in using 'gridded sea ice displacement fields' mentioned later.**

*We have expanded the introduction and appendix to better explain Lagrangian coherent structures and related Lagrangian diagnostics to identify them.*

**L 46 What characteristics of a LCS make it a hyperbolic LCS? And what makes sea ice applicable to a hyperbolic LCS? Additional arguments and descriptions of Haller et al. 2021 could be incorporated.**

*We have expanded the introduction and appendix to better explain how Lagrangian coherent structures are relevant for the study of sea ice dynamics.*

**L 50 Does the period have the 'much larger influence' or is it the identification that has it? 'Much larger influence' than what?**

*Thank you. 'Much larger influence' has been removed.*

**L 63 'the the' -> 'that the'**

*Thank you. Corrected.*

**L 63, 64 I find that this sentence does not give enough background on Haller et al. 2021 to allow for any understanding of the following equations. Please include a sentence each, with terms, on 'material stretching', 'hyperbolicity strength' and 'initial material tangent vector'. At the moment the reader is required to also read a large part of Haller et al. 2021 in order to understand these equations. It is also unclear what is represented in the two equations.**

*Thank you, we have significantly expanded the introduction and methods section to better explain our approach to quantifying sea ice dynamics.*

**L 69 My interpretation of the appendix does not show that this is "verified". If I indeed it is only likely that sea ice drift is slowly varying, then this needs to be stated as such.**

*Thank you for bringing this to our attention. The phrasing has been changed from verified to assessed.*

**L 78 including a definition of a "steady flow" will benefit this section.**

*Thank you. "(does not change with time)" has been added for clarity.*

**L 87 Again a definition of hyperbolicity in this context will aid the understanding here.**

*Thank you. This section has been changed to:*

*"TSE is positive for stretching, and negative for compression along a trajectory. Thus, if the material surrounding a buoy stretches and then compresses back to its original state, $\mathrm{TSE} = 0$. $\overline{\mathrm{TSE}}$ does not allow for this cancellation as the summand is strictly positive. It instead adds up all hyperbolic (stretching and compression) action. The term hyperbolic here comes from the dynamical systems definition of hyperbolic*

*manifolds that act as attracting and repelling structures, where nearby vectors undergo exceptional stretching or shrinking."*

**L 90 Can you add in this paragraph a description on the units of the two equations and TSE? A further description of how this relates to usual deformation units and how to interpret the two values would be of help (you may want to put this elsewhere).**

*Thank you, a discussion of units and relation to usual deformation metrics has been added to the appendix.*

**L 91 A citation is needed here.**

*Thank you. A citation has been included.*

**L 95 - 97 What is meant by this sentence? Will this technique be used later, or is it a note on the context of TSE methods and the use of existing stress vs train rheology methodologies?**

*Thank you for bringing this to our attention. This is not a technique, rather a statement about what TSE is measuring. We have clarified this further in Remark 2.*

**L 112 It is not immediately obvious why equations 6-9 are included as they are not referenced. Consider removing them. Do they apply directly to the example A3? If so put them there too. The following paragraph gives a detail dsicussion on the limitations of polygon based approaches, these extra questions don't bring anything useful here.**

*Equations 6-9 explicitly detail the calculations used for Green's theorem array-based diagnostics discussed throughout the manuscript. We have made this connection clearer by referencing the equations later. We bring up the polygon-based approaches as both the N-ICE and MOSAiC examples rely on these techniques. The Green's theorem technique is also standard for ice dynamics studies using buoys. They are the best reference to ground our new diagnostics and included as such.*

**L 123 An example of such a long time series is needed here.**

*Citations have been added.*

**L 149 An extra summary sentence here showing plain words rational for this technique would be beneficial.**

*Thank you. We have significantly changed our introduction and now explain why these datasets were chosen to validate our new approach.*

**L 169 how is the slowly varying nature of sea ice drift relate to these storm periods? Is it more or less likely that the slowly vary criterion holds?**

*The slowly varying assessment (see Appendix) is pointwise in time, so it is not related to the timescale or period of a storm.*

**L 175 I'm not sure why the beginning part of this sentence is needed, as the second part of it does not logically follow. It's fine that the diagnostic has a time window, and this description is a sensible choice.**

*Thank you. Lagrangian diagnostics are distinct from more common Eulerian diagnostics, and we are highlighting one such practical difference when comparing the findings of each.*

**L 179 here may be good place to refer to equations 6-9, if they are needed at all.**

*Thank you. This has been added.*

**L 186 a quick summary of the Itkin cleaning method would be a beneficial addition here.**

*We have rephrased this section to the following:*

*"For our analysis, we focus 24 buoy trajectories in two time windows previously examined by \citet{Itkin2017}. GPS positions were primarily sampled at 1-hour intervals, though some sampled every three hours. \citet{Itkin2017} resampled all trajectories to a 1 hr$^{-1}$ sampling frequency using a linear interpolant, and we follow this convention for our N-ICE2015 analysis. Buoy speeds that exceeded $5 km/day$ were removed and positions were resampled using a linear interpolant."*

**Figure 1. This caption requires extensive expansion. All lines need to be defined. It is currently impossible to interpret this figure without extensive reading in the text. The figure needs to be interpretable from the caption alone assuming a knowledge on the papers aims and method. Addional lines at d-1 = 0 will allow**

*Thank you. This caption has been extensively expanded.*

**L 217 - 221 Has this data been analysed by this method previously? If so citations and a summary of results is required. If not then this paper needs expansion as a presentation of these new results too. A least a discussion of previous use of these results or method is required.**

We have changed this section to the following:

*"We focus here on the paths of 101 buoys deployed within 40 km of the Polarstern. This public data set documented by \citep{Bliss2022}. The half-hourly buoy track data was cleaned following \citep{Hutchings2012}. Triads were also handpicked from the MOSAiC buoys with data spanning October 2019 to June 2020, and is the focus of a forthcoming publication. The arrays were selected to maintain reasonable shapes (no small angles, area greater than 1km$^2$) from the beginning to the end of the time series and resampled to uniform 6-hourly intervals. Handpicking triads, however, does require user discretion. Buoy tracks were resampled to match the triad sampling rate. The arrays used are shown in Figure \ref{fig:MOSAiC Array}. A deeper comparison and refinement of geometrically suitable arrays in*

*the MOSAiC data is a current topic of research. The method we use here is in line with previous work \citep{Hutchings2011, Hutchings2012}."*

**L225 please refer div, D back to the equations previously and change the labels on the plot to directly match the text. Using 2a, 2b will help too. Please also add from an improved method why TSE is compared to div, and \bar{TSE} to total D.**

*These equations are now referenced at the beginning of the N-ICE results section. We have thoroughly expanded the methods sections, as well as added an additional section in the appendix detailing the relationship between TSE, div, \bar{TSE}, and total D.*

**L 240 which source do these numbers come from? The polygon of triangle based methods? What numbers come from the other method? Can the two methods be dimensionally compared in this way? Is there any method that allows for the integral of all deformation and TSE over the period discussed?**

*These values come from the array-based methods, as explained in the new data section and at the beginning of the MOSAiC results. The values of the diagnostics are not interchangeable, as detailed in the new appendix section comparing TSE and array-based diagnostics, but dimensionally they have the same units. We could integrate the total deformation and calculate \bar{TSE} for the entire period, but that would provide us with only two scalars without a direct comparison, instead of looking for distinct temporal deformation features, as is our goal.*

**L 241 Is this value significant? Which line plot does it come from? Do all significant deformations have a higher value?**

*We have added a reference to "subplot b". We have changed the text to the following:*

*"In the 3-day window following the Apr 17 TSE and TSE peak, the mean buoy divergence oscillated around zero (Figure 6b), with the magnitude staying below 0.1d−1. This is approximately 1% of peak values of mean divergence, suggesting a relatively insignificant period of divergence. This is in contrast to TSE and TSE on April 17 which sits at approximately 50% of their total peak values, suggesting a relatively motion with a larger contribution to ice dynamics at the same time. "*

**L 242 Is shear plotted anywhere? How do we interpret shear against the TSE metrics?**

*Shear is not plotted, but can be inferred as it is loosely difference between the magnitude of divergence and total deformation. Its value is not particularly relevant for the present analysis and adding it does not reveal any additional insights, while making the figures busier.*

**Figure 2. Caption needs expanding. (a-d) are referenced in the text but do not appear. What is the black line in b and d? A scale is required for the colourbar.**

*Thank you. All captions in the manuscript have been expanded.*

**L 258 Please comment on how the spacing of the buoys and the time handling of this data (linear sampling) affects the dimensionality of the calculated TSE in comparison to high time resolution data from the other sources.**

*It is not clear to me what the reviewer is referring to with this comment. The dimension of TSE does not change, it is always a scalar value. We have however included a comment regarding the possible effects of shorter sampling periods. Linear subsampling would have the same effect on TSE as it does on decreasing the spacing the for a Riemann sum that is approximating an integral.*

**L 259 Shown in black where?**

*Thank you, we have clarified which plot we were referring to.*

**L 267 In the line plot I see that at the beginning of the period the TSE is distributed about zero, and then towards these events the spread of values reduces to oscillating peaks. Is this what you mean?**

*We have rephrased this sentence to*

*"The first event corresponds to stretching from March 26 to March 29, 2017. Previous mean TSE gradually increased built up until the absolute maximum of mean TSE on March 26."*

**L 270 Red is positive TSE? So equivalent to net divergence?**

*No, comparisons of TSE and Eulerian rate-of-strain diagnostics have now been shown in a new appendix section.*

**L271 high positive or negative values?**

*We have clarified we mean positive.*

**L 275 This paragraph will be aided by a previous discussion of what TSE we expect for certain dynamics events. For the accelerating ice described here, what TSE is expected? For constant but rotating flow as described at the end of this paragraph what idealised TSE is expected?**

*Thank you for bringing this up. We would need to do a climatic analysis of TSE values to identify expected values of TSE. This is beyond the scope of the present analysis and a topic of future research. The value of TSE is that we can locally identify significant events compared with surrounding time periods. We have however included a section in the appendix where we calculate many diagnostics for a simple analytic flow with both high shear and rotational regions.*

**Figure 3 You have chose not to include \bar{TSE} int his plot. Can you explain why? Please use a divergent colour scale for the divergence, with white at zero and different colours for positive/ negative TSE.**

*Thank you for bringing this to our attention. The color scale has been changed to a divergence color scale. \bar{TSE} is not included as it does not reveal any additional information.*

**L 281 This paper has provided not explantation I could interpret so far on how TSE provides insight in to "distant fracture events". Please expand, as this is a useful contribution if true.**

*Thank you. The following paragraph has now been added to the end of the IABP analysis:*

*"One particular benefit displayed in this example is the significant spatial extent of the large positive TSE values prior to each fracture in the Beaufort gyre. Not only was the edge of the gyre identified in the gap between positive and negative TSE, but positive TSE was also found thousands of miles from the Prince Patrick and Banks Island fractures. This supports the ability of TSE to identify the Lagrangian coherent structures in the mobile pack ice as whole, not just locally highlight a fracture."*

**L 285 I have seen the deformation events captured in the new technique, but nothing on prediction. Perhaps the wrong word to use here.**

*Thank you for bringing this to our attention. We have added the following paragraph to indicate build-up of stress that was measured by TSE prior to the major fracturing in spring 2017:*

*"The first event corresponds to stretching from March 26 to March 29, 2017. Previous mean TSE oscillations gradually increased to the absolute maximum of mean TSE on March 26. Prior to any evidence of detachment of the mobile pack ice in the Beaufort sea, TSE values were indicating an ongoing increase of stress and strain leading to the major fracturing in March and April, 2017."*

**L 291 "predicted" again this is the wrong word I think. Captured or similar is more accurate.**

*This sentence has been rephrased to more accurately reflect our findings:*

*"Approaching sea ice dynamics through quasi-objective stretching, we were able to capture coherent deformation events in concentrated buoy experiments, and even predict spring breakup in large sparsely-sampled IABP data."*

**L 296 "Buildup of stress" is this your hypothesised stress state prior to the break up, or does the TSE measure stress? Are there other measurements of stress during this period that can back up this claim?**

*There are no stress measurements available for this analysis, but we find this is a plausible explanation to the material failure that occurs after significant periods of stretching.*

**L298 Again is this a hypothesis of what internal stresses are expected within the pack? If these assessments of ice stress are speculation, please be very clear about this, or remove them.**

*Thank you, this particular speculation on stress has been removed.*

**L 317 This is the first mention of the technical methodology used in this study. Please include this information earlier in the study also. A data section detailing the exact values taken from the buoy data is needed.**

*We now have included a statement regarding the simplicity of TSE calculations in the methods section:*

*"TSE is calculated using only buoy speed and does not require projection to orthogonal velocity components as in Green's theorem approximations from arrays. Speed can be easily calculated using geodesics between GPS locations, which prevents any inconsistencies of results due to map projections. Furthermore, TSE is parameter-free with integration time being the only user-chosen value."*

*Information about the buoy data is now provided in the new data section.*

**Appendices**

**L 362-365 Can you comment on how figure A1 suggests that the slowly varying criterion is met, but not conclusively? I see from figure A1 that |vt|/|a(t)| < 1 in most cases (positive tail greater than 10^0), but not strictly or << 1. Is this correct? Does that mean that the data presented suggests that sea ice is, in general, slowly varying, but not strictly so? Can you comment on the cases where |vt|/|a(t)| > 1, when do such cases occur?**

*Your understanding of the figure is correct. Most of the ratio values are below one, but not strictly speaking. We have added a comment on when the magnitude of v(t)/a(t) is greater than unity.*

**L 394 A summary sentence for this example, repeating, and expanding upon the introduction to the appendix would be helpful here.**

*Thank you. Our appendix section has been significantly restructured and this frame-indifference violation section has been removed.*

---

## Author Comment (AC3)

Thank you for taking the time to review our manuscript. We have responded to your comments below.

General comments

**It is not completely clear to me how fractures should be captured by this diagnostic, since we do not deal with a classical stretching-compression feature in a continuum. Fractures are highlighted as identifiable features at the beginning of the results (L168, L215 and in other sentences). Shear is not necessarily assimilated to a fracture. The authors should clarify this concept from the introduction, especially because one of the examples is fully dedicated to fracture identification.**

*Thank you for bringing up this point. We have included an additional Appendix that mathematically relates trajectory stretching exponents to more classically studied rate-of-strain tensor diagnostics, including shear. There, we clarify that TSEs are not interchangeable with shear or divergence. That being said, the reviewer brings up a good point. TSEs spatially and temporally localize large stretching in the direction of the sea ice buoy trajectory. This does not guarantee a fracture either, but fracture is potential outcome from significant ice stretching and compression. We have reframed the manuscript so that this method is not a tool for identifying fracture, rather we use fractures as an external validation that TSEs have indeed identified significant stretching/compression (that actually led to factures). This is now clarified further in the abstract and throughout the text.*

**This manuscript would benefit from a more unified description of the examples, to remove any inference of cherry-picking (L147-165). This is all done in the method section, but the rationale of the choice is not discussed. There is a major focus on the role of storms in setting ice conditions, but the choice of the examples is more varied, especially with the inclusion of fractures. This diversity is appreciable but may be confusing, since there is an expectation that this diagnostic application is to identify the presence of synoptic events. This seems to be the case in the first two examples, but then it is not summarised in the discussion/conclusion. The authors may consider to better frame the applicability of the method with a more general introduction**

*Thank you for the suggestion. The manuscript has been re-written and re-organized with a section devoted to the different experimental data sets. We have included a more general introduction and explanation of our intents. We have also further clarified this method is not fracture specific, nor is it synoptic event specific, but we are using synoptic events as they provide a useful timescale at which we can verify our identification of stretching/compression through storm analysis and remote sensing comparisons. We have improved the text to reflect this.*

**The authors have made available the code that should putatively reproduce the results presented in this manuscript. However, this is not attainable. The code is the same referenced by Haller et al. (2021), which was meant to compute TSE for ocean applications. It cannot be used to compute the TSE for sea-ice buoys. In collaboration with a PhD student in the group, we implemented the numerical computation of TSE for individual buoys from eq. 4-5, and we found some ambiguities in the choice of the discretization stencil that would affect the results. This is normal with numerical**

**discretizations, but as it stands, a reader would not be able to implement the method and obtain the same results.**

*Thank you for bringing this to our attention. We have now provided an example script "TSE_Buoy_Example.m" in the same repository, specifically for calculating TSEs from buoy trajectories. This function only requires latitude, longitude, and timestamp vectors. This script assumes a uniform temporal sampling as the GPS data were all resampled to uniform samplings in the three examples in the text, but leaves the actual discretization of the data up to the user's preference.*

**This method is alternative and superior to the use of buoy clusters and polygons. This is clearly demonstrated in the results and appendices, but not explained in the introduction. The Method section is somewhat explained the other way around, with the existing methods described at the end, but invoked earlier. I honestly struggled to follow it, and I would recommend some restructuring, especially for readers who are not fully aware of the underlying mathematical concepts.**

*Thank you. We have reorganized the manuscript to improve the flow and clarity of the topics and included an expository appendix section deriving TSEs.*

**Following up from the previous point, my main question is how different this method is from the single particle dispersion applied in Rampal et al. (2009) and other literature referenced in the manuscript. I am not sure this is addressed in the manuscript. The authors state at L98 that there are limited Lagrangian alternatives to compare to, but this comparison is not presented.**

*We have clarified the difference between our Lagrangian approach and the Rampal et al. (2009) dispersion rates. Pairwise dispersion relies on pairs of buoys, and is not a single buoy analysis like TSE. Their approach is a slightly modified version of relative dispersion commonly used in oceanography, with which TSEs have already been compared in previous studies. Rampal et al. also rely on manipulations and assumptions that we do not. This is also referenced in the new text.*

**The introduction to the Result section at L176-L182 is quite problematic and needs a thorough revision. These paragraphs are more akin to the Method section. The choice of the frequency of analysis is based on the synoptic scale, but then the same method is applied to the last example involving fractures, which may be related to internal ice stresses rather than storms.**

*Thank you. The introduction to the results section have been changed and the referenced paragraphs have been moved to the methods section. We further explain our choice of Lagrangian timescale, and show the influence of variability in that choice.*

**There is no sensitivity analysis on how TSE is affected by the choice of the sampling window, as well as the granularity of the source data. For instance, the authors say they have linearly interpolated to hourly frequency, but there is no justification for this choice. This is especially important when using the IABP data that have highly varying frequencies.**

*Thank you for bringing this to our attention. We now include a comparison with different TSE time scales in the Appendix. We have provided a better explanation of the choice of sampling frequency. We now explain how these choices were based on the datasets being analyzed, and other studies on the topic.*

**I have some more specific points related to this section that should be addressed in the revised method section**

**Not clear how the 3-day window would "balance the high temporal resolution of TSE" (artificial, since it is linearly interpolated to 1 hr), "while dampening influence of measurement noise and sub-daily oscillations". Maybe it's just the English, but I do not understand what the authors mean. There are known sub-daily oscillations and they will be captured in the Lagrangian estimation of velocity (Gimbel et al., 2012)**

*Thank you, we have clarified and rewritten this section in connection with our new section on integration periods in the appendix.*

**Noise is mentioned several times in these paragraphs but never quantified (also in results, e.g. L207). What do the authors mean by noise? Inertial oscillations are not noise, they are signal**

*Thank you. We have clarified what we mean by noise. There are clear artificial influences in some of the gps signals that cannot be attributed to inertial oscillations.*

**I do not understand why TSE should always precede significant storm events, and why this should depend on the choice of the 3-day window. Indeed, in Fig. 1 there are few cases in which TSE This is maybe where having the code or showing the discretization of the TSE computation would help.**

*Thank you. As mentioned above, we have provided a new code. TSE is a Lagrangian diagnostic evaluated over a certain length of a trajectory. To generate a TSE time series, a summation of instantaneous values is performed in a forward-looking fashion so that the TSE timeseries appears predictive. This is by design from dynamical systems. We have further explained this in the methods section.*

**My understanding is that the TSE is computed over a rolling window, so, as long as the window is not larger than the scale of a storm (up to 3-5 days), it would detect the feature. This argument is used throughout the presentation of the results (e.g. L196) but not made fully explicit.**

*As TSE is computer for a specific section of a buoy trajectory, you are measuring stretching for that section. If the time window is larger than the scale of a storm, you will still be measuring all the stretching during the storm, but possibly include quiescent periods before and after the storm. This could result in a lower TSE value as we are normalizing by integration time, but this does not mean we cannot detect features at time scales smaller (or larger) than the window of computation. We have explained this further in the methods.*

**Also, it is not clear how a storm is defined and shown in Fig. 1 (when the core of the cyclone is the closest to the buoy location? E.g. Vichi et al., 2019, for an example from**

**Antarctic sea ice, or maybe when the MSLP is lower than a certain threshold). My question is whether the authors think the stretching-compression is enhanced when the storms approach the buoy location. And, maybe, after the passage (as reported by Itkin et al., 2017), due to wave-induced breaking, sea ice goes into free-drif state which indicates weaker LCS. The authors should make an effort to interpret this important feature.**

*Thank you. The storm definition came from previous N-ICE evaluations, including meteorological station data. This was not performed by us, and is referenced in the text, as is the subsequent analysis of sea-ice response by Itkin. We have expanded on the IABP example where the sea ice is transition towards a free-drift state to better explain these features.*

**The authors state they have made a sensitivity analysis on this (L178-179) but this is not presented in the results**

*Thank you for bringing this miscommunication to our attention. The cross-correlation you are referring to is not a sensitivity analysis, rather a quick validation of what is qualitatively presented. We do not think this computation warrants further validation as it is an observation supported by all the TSE peaks that precede stretching events discussed at length in the results.*

**Finally, but this is just a minor point, I would advise the authors to briefly discuss the possible application of this method in Antarctic sea ice, and maybe give a recommendation on what would be the best approach.**

*Thank you for the suggestion. We have added a small discussion on Antarctic sea ice. This is a venue some of the authors have been investigating, and may prove to benefit from TSE calculations.*

**Specific comments**

**L35-37 This sentence requires some references. These references come later in the manuscript, but I think a brief introduction on the Lagrangian coherent structures would be of aid to the sea-ice experts that are less familiar with them. LCS are more common in ocean dynamics and less in the sea ice.**

*Thank you. We have modified the introduction to improve clarity and provide more references.*

**L42-49 This paragraph relies on the previously published papers by Haller et al. I acknowledge the value of those papers to provide the mathematical background for this application. They may not be entirely approachable by a variety of scientists interested in applying this diagnostic further, possibly not noticing the limits of applicability. The authors make the implicit assumption that they are directly applicable to sea ice. I am aware that this is partly addressed later and in Appendix A1 (see my comments below), but I would suggest an earlier introduction to the concept.**

*Thank you again for bringing this up. We seek to have this work accessible to a wide range of scientists and as such we have modified this introduction section to improve clarity and provide more references.*

**L57 and L77: some ambiguities in the use of the notation. Is the trajectory symbol in italic and bold? Than it should be consistent throughout the manuscript**

*Thank you. This has been corrected.*

**L69-72 This is a major assumption, and since it has been verified in the realm of ocean currents in the cited paper, it is likely to be acceptable with sea ice drift. However, current detection from space is more accurate and less prone to the resolution issue of sea-ice drift retrieval. I would recommend the authors to bring back this issue of the . I also have a few issues with the "verification" in A1, which, given the many uncertainties, I would rather call this process "assessment of the main assumption". The choice of the period will define the maximum speed of the Lagrangian velocity. There is no explanation in A1 on how the Lagrangian velocity has been computed, nor on the period used for this analysis. In the caption of Fig. A1 it only says "50 days of sea ice trajectories in 2017". This distribution will certainly change with different years, regions, etc. None of this is included in the presented analysis.**

*We have changed from verification to assessment and discussed the connection between the 50-day evaluation window and the IABP experiment. We have significantly expanded on the influence of slowly-varying conditions and how TSEs are derived using this assumption, both in the text and in a new appendix section.*

**L78-79 I would suggest the authors to reiterate the concept of quasi-objective diagnostics at this point of the manuscript**

*Thank you, this has been clarified.*

**L89 Eq 7-9 have not been introduced yet, as well as the concept that this method is alternative to the use of buoy clusters and polygons. I would suggest the authors make clear from the beginning of the Method section that there are existing alternatives and this is complementary to them.**

*Thank you. We have reorganized the development of the polygonal and single-buoy methods.*

**L97 Maybe "nature" is missing in this sentence**

*Nature has been added. Thank you.*

**Eq. 6 These equations are presented in discretized notation, but this is not done for the TSE. I understand that this method is more classical and it is somewhat obvious, but it would still require some definition of what u and v are. I noticed it when discussing with MSc and Phd students that struggled to understand the notation.**

*Thank you for pointing this out. These equations have been modified for clarity.*

**L124-125 Any method with more than one buoy, or with buoys covering a larger regional expanse where constellations are less reliable, will be affected by the signal-to-noise ratio. I understand this may be more of relevance to the Southern Hemisphere.**

*Thank you for this clarification.*

**Fig. 1 Please explain what the dashed line in panels a and c represents. I would also recommend to use the same range in the Y axis of panels b and d, to better compare late winter with spring.**

*Thank you for bringing this to our attention. We reference values exceeding 1d^{-1} in the text, but neglected to connect this with the dashed lines. This has now been mitigated in the text and in the caption. We have rescaled the y-axes as well.*

**L212 Please report the frequency of the MOSAiC buoys and if the same**

*This has been added to the text.*

**L225 January 14 is not very visible in the figure with the current choice of ticks. Also, the authors state that it is more extreme in TSE. More extreme than what?**

*Thank you, this figure and comparison has been improved.*

**L236 this reference to Copernicus data has a non-existent DOI**

*Thank you. We have changed this reference to fit the standards of ASF and ESA (i.e. https://asf.alaska.edu/data-sets/sar-data-sets/sentinel-1/sentinel-1-how-to-cite/)*

**L220-239 The April example has been chosen because of SAR images available before and after the event. Maybe this could be mentioned at the beginning of the paragraph. The interpretation of the coloured points is not given in this section, and the reader is left with a sense of incompleteness. Also, there are no letters in the panels of Fig 2 and the colormap choice does not clearly show negative and positive values. This latter comment applies to all the figures with colorbars. This colormap is not colorblind friendly.**

*Thank you. This before/after has been clarified. As well, we have changed the colormaps to a divergent colorscale when indicating positive vs negative values, and sought colorblind friendly linear maps.*

**L242 I am not sure LKF is spelt out in the text. Also this sentence is quite obscure. Does it mean that the subjective choice of the clusters (L217) would change the results?**

*LKF is spelt out in the introductory paragraph of the new dataset section. Yes, the subjective nature of cluster choice does influence what is detected.*

**L246 Please explain the meaning of "previously neglected periods"**

*Thank you. This sentence has been expanded to*

*"In this scenario, the stretching and relaxation measured by TSE presents a clear correlation with material deformation of the ice and suggests TSEs may provide ice behavior insight during times when Green's theorem methods are not possible, such as when there are too few buoys or they are their orientation is inappropriate, and when array-based approaches have underestimated dynamic behavior."*

**Sec 3.5 Please indicate how many buoys have been used, what is the range of sampling frequency and why the interpolation frequency is now 6-hourly. I wonder if the 3-day window is justified in this context, and if yes, it should be justified. LKF are rather random events, not necessarily linked to the synoptic scales**

*We have further explained the sampling frequency and the Lagrangian calculation window in the new dataset and methods sections.*

**L269-270 Please explain what previous behaviour means here. This does not seem to be shown. Only points from the peak period are presented. The fact that TSE is positive and apparently saturated, it may mean that the event has already started, but this is not clear.**

*We have changed this sentence to the following:*

*"All buoys in the free-drift region have high (red) TSE values and create another local maximum in the mean time series, further supporting this relatively significant stretching event in March and April when compared to times prior to and after these months."*

**L271 Bank Islands**

*Capitalization Corrected. Thank you*

**L271-272 The buoys in the south show a clear stretching-compression cycle during this period but no evident fracturing. Maybe the authors can comment further on what kind of feature is being detected by the method, and whether it is realistic.**

*Thank you. We have further expanded on the difference of buoys inside the free-drift and outside the free-drift zone.*

**L300-303 The English can be improved. I also think this is a rather bold statement, given that this diagnostic is only related to sea ice dynamics. It can be associated with other data to obtain further insights in the coupling. The points after this sentence do not critically assess the required improvements as indicated in the sentence.**

*Thank you. We have rephrased this statement as follows*

*"The single-buoy quasi-objective trajectory stretching exponents (TSEs) identify dynamic sea ice events that are potentially significant in terms of understanding spatially and temporally varying sea ice deformation. As sea ice dynamics plays an important role in atmosphere-ice-ocean exchange processes, we find the further event-detection sensitivities possible with TSEs*

*are a valuable complement to common, polygon-based divergence, shear, and deformation approximations."*

**L314 Correlations? Do the authors mean approximations?**

*Thank you for the question. TSEs do not measure fractures or break-up events, and thus the high trajectory stretching exponents are only correlated with the concurrent fractures and break up that we observed.*

**L331 of TSE signals**

*Thank you. Corrected.*

**L336 I would argue with this statement. I think this is what the methodology would allow. The results show very promising applications of this method, but the interpretations are still in a preliminary phase.**

Thank you. We have improved this line to reflect your understanding.

**Figures in Appendix: they should all be renumbered (not S2 but A2).**

*The figures have been renumbered.*

Appendix A2

**Please explain if the rotation is done with the same angle. I would suggest first to discuss the flow field, and shift panel A2c to panel A2a. The buoy locations could also be added on that field, and the reader would see that they are meant to approximate the whole field and not local regions. I would also suggest to limit the X-axis of A2b to the range between 0 and 50, to make it more realistic. The convergence is indeed rather rapid, but it is not clear where it does happen**

*Thank you for the suggestions. Figure A2 has been adapted following your suggestions.*

Appendix A3

**This is another excellent example, but it is not adequately generalised in the text. The point is very well made, but the implications are not completely clear to the reader less interested in the mathematical formulation. The chosen flow is rather peculiar (a locally divergent flow, probably less relevant in sea-ice dynamics) and one may argue that this conclusion cannot be generalised to any kind of flow.**

*Thank you for bringing this to our attention. We have removed this example from the appendices as we found it brought more confusion to the reader than clarity.*

References

Gimbert, F., Marsan, D., Weiss, J., Jourdain, N.C., Barnier, B., 2012. Sea ice inertial oscillations in the Arctic Basin. The Cryosphere 6, 1187–1201. https://doi.org/10.5194/tc-6-1187-2012

Vichi, M., Eayrs, C., Alberello, A., Bekker, A., Bennetts, L., Holland, D., de Jong, E., Joubert, W., MacHutchon, K., Messori, G., Mojica, J.F., Onorato, M., Saunders, C., Skatulla, S., Toffoli, A., 2019. Effects of an Explosive Polar Cyclone Crossing the Antarctic Marginal Ice Zone. Geophysical Research Letters 2019GL082457. https://doi.org/10.1029/2019GL082457